# Statistical Analysis of the Sinkhorn Iterations for Two-Sample Schrödinger Bridge Estimation

**Ibuki Maeda**[1]
maeda.ibuki554@mail.kyutech.jp

**Rentian Yao**[2]
rentian2@math.ubc.ca

**Atsushi Nitanda**[3,4,5]
atsushi_nitanda@a-star.edu.sg

[1] Department of Artificial Intelligence, Kyushu Institute of Technology, Japan
[2] Department of Mathematics, University of British Columbia, Canada
[3] Institute of High Performance Computing, Agency for Science, Technology and Research (A⋆STAR), Singapore
[4] Centre for Frontier AI Research, Agency for Science, Technology and Research (A⋆STAR), Singapore
[5] College of Computing and Data Science, Nanyang Technological University, Singapore

## Abstract

The Schrödinger bridge problem seeks the optimal stochastic process that connects two given probability distributions with minimal energy modification. While the Sinkhorn algorithm is widely used to solve the static optimal transport problem, a recent work (Pooladian and Niles-Weed, 2024) proposed the *Sinkhorn bridge*, which estimates Schrödinger bridges by plugging optimal transport into the time-dependent drifts of SDEs, with statistical guarantees in the one-sample estimation setting where the true source distribution is fully accessible. In this work, to further justify this method, we study the statistical performance of intermediate Sinkhorn iterations in the two-sample estimation setting, where only finite samples from both source and target distributions are available. Specifically, we establish a statistical bound on the squared total variation error of Sinkhorn bridge iterations: $\mathcal{O}(1/m + 1/n + r^{4k})$ ($r \in (0, 1)$), where $m$ and $n$ are the sample sizes from the source and target distributions, respectively, and $k$ is the number of Sinkhorn iterations. This result provides a theoretical guarantee for the finite-sample performance of the Schrödinger bridge estimator and offers practical guidance for selecting sample sizes and the number of Sinkhorn iterations. Notably, our theoretical results apply to several representative methods such as [SF]$^2$M, DSBM-IMF, BM2, and lightSB(-M) under specific settings, through the previously unnoticed connection between these estimators.

## 1 Introduction

In recent years, there has been increasing interest in Schrödinger bridge problems, with applications in mathematical biology (Chizat et al., 2022; Lavenant et al., 2024; Yao et al., 2025), Bayesian posterior sampling (Heng et al., 2024), generative modeling (De Bortoli et al., 2021; Wang et al., 2021), among others. The Schrödinger bridge problem aims to minimize the Kullback–Leibler (KL) divergence with respect to the Wiener measure, the distribution of Brownian motion over the path space, given two fixed marginal distributions $\mu, \nu$ as source and target distributions.

The Schrödinger Bridge problem is also strongly connected to the entropic optimal transport (EOT) problem, which offers an efficient and regularized alternative to classical optimal transport, and is solvable via the Sinkhorn algorithm (Cuturi, 2013). The Sinkhorn algorithm operates in the dual space, optimizing the Schrödinger potentials between empirical distributions, and enjoys well-established convergence guarantees (Franklin and Lorenz, 1989; Peyré and Cuturi, 2018). Recently, Pooladian and Niles-Weed (2024) introduced the Sinkhorn bridge method, which plugs in the optimal Schrödinger potentials into the time-dependent drifts of stochastic differential equations to estimate

the Schrödinger bridge between the source and target distributions $\mu$ and $\nu$. Their analysis established statistical bounds, adaptive to the intrinsic dimension of the target distribution $\nu$, in the one-sample estimation setting, where the true source distribution $\mu$ is fully known.

## 1.1 Contributions

In this work, to better understand the algorithmic and statistical convergence rates of the Sinkhorn bridge method, we study the statistical performance of intermediate estimators obtained by Sinkhorn iterations in the two-sample estimation setting, where only empirical distributions $\mu_m$ and $\nu_n$ of $m$- and $n$-finite samples from $\mu$ and $\nu$ are available.

**Statistical guarantee for Sinkhorn bridge iterations.** First, we establish the following statistical bound (Theorem 1) for the Sinkhorn bridge estimator by leveraging analytical tools for the Schrödinger bridge in the one-sample setting (Pooladian and Niles-Weed, 2024) and for entropic optimal transport (Stromme, 2023b):

$$\mathbb{E}\left[\mathbf{TV}^2(P_{\infty,\infty}^{*,[0,\tau]}, P_{m,n}^{*,[0,\tau]})\right] = \mathcal{O}\left(\frac{1}{m} + \frac{1}{n}\right), \tag{1}$$

where $P_{\infty,\infty}^*$ and $P_{m,n}^*$ denote the true Schrödinger bridge and its Sinkhorn bridge estimator based on the optimal EOT solution, and $P_{\infty,\infty}^{*,[0,\tau]}$ and $P_{m,n}^{*,[0,\tau]}$ are restrictions of them on the time-interval $[0,\tau]$. The result improves upon the bound established in (Pooladian and Niles-Weed, 2024) by extending the analysis to the two-sample setting and sharpening the rate in $n$ from $n^{-1/2}$ to $n^{-1}$ at the cost of a deterioration in the dependence on $\varepsilon$ and $R$, where $\varepsilon$ is the strength of entropic regularization and $R$ is the radius of the data supports.

Next, we analyze the intermediate iterations of the Sinkhorn algorithm and prove a convergence rate (Theorem 2) for the corresponding path measures:

$$\mathbb{E}[\mathbf{TV}^2(P_{m,n}^{*,[0,\tau]}, P_{m,n}^{k,[0,\tau]})] \lesssim \frac{\tau}{1-\tau}\frac{R^4}{\varepsilon}\left(\tanh\left(\frac{R^2}{\varepsilon}\right)\right)^{4k}, \tag{2}$$

where $P_{m,n}^{k,[0,\tau]}$ denotes the Sinkhorn bridge estimator at the $k$-th Sinkhorn iteration for $\mu_m$ and $\nu_n$. Together, these results imply the convergence rate $\mathbb{E}[\mathbf{TV}^2(P_{\infty,\infty}^{*,[0,\tau]}, P_{m,n}^{k,[0,\tau]})] = \mathcal{O}(1/m + 1/n + r^{4k})$ $(r \in (0,1))$ regarding the number of samples $m$ and $n$, and the number of Sinkhorn iterations $k$.

**Relationship with the other Schrödinger Bridge estimators.** Remarkably, our theoretical results on the Sinkhorn bridge can be directly applied to the representative Schrödinger Bridge solvers: [SF]$^2$M (Tong et al., 2023), DSBM–IMF (Shi et al., 2023; Peluchetti, 2023), BM2 (Peluchetti, 2024), and lightSB(-M) (Korotin et al., 2023; Gushchin et al., 2024). First, the optimal estimator produced by these methods coincide with the Sinkhorn bridge estimator (Proposition 1). Moreover, when DSBM-IMF and BM2 are initialized with the reference process, the estimator after $k$ iterations matches the Sinkhorn bridge obtained after $k$ iterations of the Sinkhorn algorithm (Proposition 2). Furthermore, Algorithm 1, which learns the Sinkhorn bridge drift via a neural network, is a special case of the procedure proposed in [SF]$^2$M. Consequently, our generalization-error analysis applies to all of these methods, endowing the algorithms with theoretical guarantees.

## 1.2 Related Work

**Schrödinger bridge problem.** The connection between entropic optimal transport and the Schrödinger bridge (SB) problem has been extensively studied; for a comprehensive overview, see the survey by Léonard (2013). More recently, there has been growing interest in computational approaches to the SB problem, particularly those leveraging deep learning techniques (De Bortoli et al., 2023; Shi et al., 2022; Bunne et al., 2023; Tong et al., 2023). In parallel, several studies have explored classical statistical methods for estimating the SB (Bernton et al., 2019; Pavon et al., 2021; Vargas et al., 2021). For example, Bernton et al. (2019) proposed a sampling framework based on trajectory refinement via approximate dynamic programming. Pavon et al. (2021) and Vargas et al. (2021) introduced methods that directly estimate intermediate densities using maximum likelihood principles. Specifically, Pavon et al. (2021) presented a scheme that explicitly models the target

density and updates weights accordingly, while Vargas et al. (2021) estimated forward and backward drifts using Gaussian processes, optimized via a likelihood-based objective. Assuming full access to the source distribution, the statistical performance of the Schrödinger bridge estimator was analyzed in the one-sample setting by Pooladian and Niles-Weed (2024). Other estimators, such as those based on neural approximations or efficient algorithmic variants, have been evaluated in recent works (Korotin et al., 2023; Stromme, 2023a).

**Entropic optimal transport.** Entropic Optimal Transport (EOT) introduces an entropy term to the standard Optimal Transport (OT) problem (Cuturi, 2013), primarily proposed to improve computational efficiency and smooth the transport plan. This regularization makes the OT problem strictly convex, enabling the application of the Sinkhorn algorithm, an efficient iterative method for the dual problem (Cuturi, 2013). The introduction of the Sinkhorn algorithm has allowed the OT problem, previously computationally very expensive, to obtain approximate solutions in realistic time even for large-scale datasets, leading to a dramatic expansion of its applications in the field of machine learning. For example, there are diverse applications such as measuring distances between distributions in generative modeling (Genevay et al., 2018), and shape matching or color transfer in computer graphics and computer vision (Solomon et al., 2015). The theoretical aspects of EOT have also been deeply studied. Analyses regarding the convergence rate of the Sinkhorn algorithm (Franklin and Lorenz, 1989) and the statistical properties (such as sample complexity) when estimating EOT from finite samples (Genevay et al., 2019) have been established. Furthermore, EOT is known to have a close relationship with the Schrödinger Bridge (SB) problem (Léonard, 2013). EOT in a static setting is equivalent to the problem of matching the marginal distributions of the SB problem, which deals with dynamic stochastic processes, and both share common mathematical structures in duality and optimization algorithms. This connection suggests that theoretical and computational advances in one field can contribute to the other.

## 1.3 Notations

For a metric space $\mathcal{X}$, let $\mathcal{P}(\mathcal{X})$ be the space of probability distributions over $\mathcal{X}$, and $\mathcal{P}_2(\mathcal{X})$ be the subset of $\mathcal{P}(\mathcal{X})$ with finite second-order moment. Let $\mathbb{R}^d$ be the $d$-dimensional Euclidean space and $B(a, R) = \{x \mid \|x - a\|_2 \leq R\} \subset \mathbb{R}^d$ be the Euclidean ball of radius $R > 0$ centered at $a \in \mathbb{R}^d$. For real numbers $a$ and $b$, we set $a \vee b = \max(a, b)$ and $a \wedge b = \min(a, b)$. We write $a \lesssim b$ to denote that there exists a constant $C > 0$, independent to $a$ and $b$, such that $a \leq C b$. We denote by $\delta_x$ the Dirac measure concentrated at the point $x \in \mathbb{R}^d$. We denote by $\mathbb{1}_d \in \mathbb{R}^d$ the $d$-dimensional vector whose components are all equal to one. For a measure $\mu$ and a function $f$, we write $\mu(f) = \int f(x) \, \mu(dx)$. Let $\mathrm{TV}(\mu, \nu)$ be the total variation distance between two probability distributions $\mu$ and $\nu$. For $\varepsilon > 0$ and any real number or real-valued function $h$, write $\exp_\varepsilon(h) := \exp(h/\varepsilon)$. We denote the $d$-dimensional probability simplex by $\Delta_d := \{\alpha \in \mathbb{R}_{\geq 0}^d \mid \sum_{i=1}^d \alpha_i = 1\}$.

## 2 Preliminaries

### 2.1 Entropic Optimal Transport

Let $\mu, \nu \in \mathcal{P}_2(\mathbb{R}^d)$ and $\varepsilon > 0$ be fixed. The entropic optimal transport (EOT) cost between $\mu$ and $\nu$ is

$$\mathrm{OT}_\varepsilon(\mu, \nu) := \inf_{\pi \in \Pi(\mu,\nu)} \left\{ \int_{\mathbb{R}^d \times \mathbb{R}^d} c(x, y) \, \mathrm{d}\pi(x, y) + \varepsilon \, H(\pi \,\|\, \mu \otimes \nu) \right\}, \tag{3}$$

with the cost function $c(x, y) := \frac{1}{2} \|x - y\|_2^2$. Here, $\Pi(\mu, \nu) \subset \mathcal{P}_2(\mathbb{R}^d \times \mathbb{R}^d)$ denotes the set of couplings with marginals $\mu$ and $\nu$, and

$$H(\pi \,\|\, \mu \otimes \nu) := \int_{\mathbb{R}^d \times \mathbb{R}^d} \log\left(\frac{\mathrm{d}\pi}{\mathrm{d}(\mu \otimes \nu)}(x, y)\right) \mathrm{d}\pi(x, y)$$

is the Kullback–Leibler divergence between $\pi$ and $\mu \otimes \nu$. If $\pi$ is not absolutely continuous with respect to $\mu \otimes \nu$, we set $H(\pi \,\|\, \mu \otimes \nu) = +\infty$.

The optimization problem in Eq. (3) is strictly convex with respect to $\pi$, and thus it admits a unique solution $\pi^*_{\infty,\infty} \in \Pi(\mu,\nu)$. Furthermore, the problem has a dual formulation given by

$$\mathrm{OT}_\varepsilon(\mu,\nu) = \sup_{f \in L^1(\mu), g \in L^1(\nu)} \Phi^{\mu\nu}(f,g), \quad \text{where}$$

$$\Phi^{\mu\nu}(f,g) := \int f \, \mathrm{d}\mu + \int g \, \mathrm{d}\nu - \varepsilon \int \left( \exp_\varepsilon \left( f(x) + g(y) - \frac{1}{2}\|x-y\|_2^2 \right) - 1 \right) \mathrm{d}\mu(x) \, \mathrm{d}\nu(y).$$
(4)

The dual problem (4) admits the Schrödinger potentials $f^*_{\infty,\infty}$ and $g^*_{\infty,\infty}$ as optimal solutions, and the first-order optimality condition implies

$$f^*_{\infty,\infty}(x) = -\varepsilon \log \int_{\mathbb{R}^d} \exp_\varepsilon \left( g^*_{\infty,\infty}(y) - \frac{1}{2}\|x-y\|_2^2 \right) \mathrm{d}\nu(y),$$
(5a)

$$g^*_{\infty,\infty}(y) = -\varepsilon \log \int_{\mathbb{R}^d} \exp_\varepsilon \left( f^*_{\infty,\infty}(x) - \frac{1}{2}\|x-y\|_2^2 \right) \mathrm{d}\mu(x),$$
(5b)

which is known as Schrödinger system. Note that the solution is constant-shift invariant, i.e. for any $a \in \mathbb{R}$, $(f^*_{\infty,\infty} + a, g^*_{\infty,\infty} - a)$ is also a solution to the Schrödinger system (5). It can be shown that with a constraint $\nu(g^*_{\infty,\infty}) = 0$, the solution to (5) is unique. The optimal solution $\pi^*_{\infty,\infty}$ of the primal problem (3) can be expressed using the dual solution via

$$p^*_{\infty,\infty}(x,y) := \frac{\mathrm{d}\pi^*_{\infty,\infty}}{\mathrm{d}(\mu \otimes \nu)}(x,y) = \exp_\varepsilon \left( f^*_{\infty,\infty}(x) + g^*_{\infty,\infty}(y) - \frac{1}{2}\|x-y\|_2^2 \right).$$
(6)

We refer to (Nutz, 2021) for more details of entropic optimal transport problems.

## 2.2 Schrödinger Bridge

Let $\Omega = \mathcal{C}([0,1];\mathbb{R}^d)$ be the path space consisting of all continuous maps from the time interval $[0,1]$ to $\mathbb{R}^d$. Let $W^\varepsilon$ be the law of a reversible Brownian motion on $\mathbb{R}^d$ with volatility $\varepsilon$. The Schrödinger Bridge problem is then defined through the following entropy minimization problem:

$$\inf_{P \in \mathcal{P}(\Omega)} \varepsilon H(P \mid W^\varepsilon) \quad \text{subject to} \quad P_0 = \mu, \quad P_1 = \nu,$$
(7)

where $P_t \in \mathcal{P}(\mathbb{R}^d)$ is the marginal distribution at time $t \in [0,1]$ of the path-space distribution $P$, i.e. $X_t \sim P_t$ if $X$ is a stochastic process with distribution $P$. Though $W^\varepsilon$ is an unbounded positive measure over $\Omega$, the above optimization problem is still well defined as illustrated by Léonard (2013).

The Schrödinger Bridge problem is closely connected with the EOT problem. Precisely, the optimal solution $P^*_{\infty,\infty} \in \mathcal{P}(\Omega)$ of Eq. (7) can be derived by first solving the EOT problem (3) and then connect the optimal coupling with the Brownian bridge $W^\varepsilon_{|x_0,x_1} = W^\varepsilon(\cdot \mid X_0 = x_0, X_1 = x_1)$, i.e.,

$$P^*_{\infty,\infty} = \int W^\varepsilon_{|x_0,x_1} \, \mathrm{d}\pi^*_{\infty,\infty}(x_0,x_1).$$
(8)

The optimal solution $P^*_{\infty,\infty}$ can also be characterized as the path measure of the SDE defined by

$$dX_t = b^*_{\infty,\infty}(X_t,t) \, \mathrm{d}t + \sqrt{\varepsilon} \, \mathrm{d}B_t, \quad X_0 \sim \mu,$$

where $B_t$ is the standard Brownian motion, and the drift coefficient is given by

$$b^*_{\infty,\infty}(z,t) = \frac{1}{1-t} \left( -z + \frac{\mathbb{E}_{Y \sim \nu}[\gamma^{*,t}_{\infty,\infty}(Y,z)Y]}{\mathbb{E}_{Y \sim \nu}[\gamma^{*,t}_{\infty,\infty}(Y,z)]} \right)$$
(9)

with $\gamma^{*,t}_{\infty,\infty}(y,z) = \exp_\varepsilon(g^*_{\infty,\infty}(y) - \frac{1}{2(1-t)}\|y-z\|_2^2)$. We refer to Chen et al. (2021); Léonard (2012, 2013) for more details of the Schrödinger Bridge.

## 2.3 Sinkhorn Algorithm

Given the empirical distributions $\mu_m = \frac{1}{m}\sum_{i=1}^{m}\delta_{X_i}$ and $\nu_n = \frac{1}{n}\sum_{j=1}^{n}\delta_{Y_j}$, we consider the empirical EOT problem $\mathrm{OT}_\varepsilon(\mu_m, \nu_n)$. Let $f_{m,n}^*$ and $g_{m,n}^*$ be the optimal Schrödinger potentials. The Sinkhorn algorithm is an iterative method for approximating these optimal potentials by alternately updating the dual potentials $f_{m,n}^{(k)}$ and $g_{m,n}^{(k)}$ as follows.

$$f_{m,n}^{(k+1)} = \operatorname*{argmax}_{f\in L^1(\mu_m)} \Phi^{\mu_m\nu_n}(f, g_{m,n}^{(k)}), \quad g_{m,n}^{(k+1)} = \operatorname*{argmax}_{g\in L^1(\nu_n)} \Phi^{\mu_m\nu_n}(f_{m,n}^{(k+1)}, g). \tag{10}$$

We note that $f_{m,n}^{(k)}$, $g_{m,n}^{(k)}$ can be interpreted as $m$- and $n$-dimensional vectors, respectively. For efficient computation, it is common to rewrite these updates in an alternative form. Let the kernel matrix $K \in \mathbb{R}^{m\times n}$ be defined as $K_{ij} = \exp_\varepsilon(-\frac{1}{2}\|X_i - Y_j\|_2^2)$, and denote the vectors $u^{(k)} \in \mathbb{R}^m$ and $v^{(k)} \in \mathbb{R}^n$ by $u_i^{(k)} = \exp_\varepsilon(f^{(k)}(X_i))$ and $v^{(k)}(Y_j) = \exp_\varepsilon(g^{(k)}(Y_j))$. Then the update rule (10) becomes

$$u^{(k+1)} = m\mathbf{1}_m \oslash (K\,v^{(k)}), \quad v^{(k+1)} = n\mathbf{1}_n \oslash (K^T\,u^{(k+1)}),$$

where $\oslash$ denotes element-wise division [1]. This reformulation greatly improves computational efficiency and enables large-scale applications.

Let $d_{\mathrm{Hilb}}$ be the Hilbert (pseudo-)metric on $\mathbb{R}_+^d$ defined by

$$d_{\mathrm{Hilb}}(u, u') = \log\left(\max_i \frac{u_i}{u_i'} \cdot \max_i \frac{u_i'}{u_i}\right), \quad \forall\, u, v \in \mathbb{R}_+^d.$$

It is obvious that $d_{\mathrm{Hilb}}(x, y) = 0$ if and only if there exists $\lambda \in \mathbb{R}_+$ such that $x = \lambda y$. Let the initial values be $(u^{(0)}, v^{(0)}) = (\mathbb{1}_m, \mathbb{1}_n)$. Denote by $(u^{(k)}, v^{(k)})$ the solution obtained after $k$ iterations of the Sinkhorn algorithm (10), and by $(u^*, v^*) = (\exp(f_{m,n}^*/\varepsilon), \exp(g_{m,n}^*/\varepsilon))$ the optimal solution. Then, it is shown that (Franklin and Lorenz, 1989; Peyré and Cuturi, 2018),

$$d_{\mathrm{Hilb}}(u^{(k)}, u^*) \le \lambda(K)^{2k-1}\, d_{\mathrm{Hilb}}(v^{(0)}, v^*), \quad d_{\mathrm{Hilb}}(v^{(k)}, v^*) \le \lambda(K)^{2k}\, d_{\mathrm{Hilb}}(v^{(0)}, v^*).$$

Here, $\lambda(K) < 1$ is defined as $\lambda(K) := \frac{\sqrt{\gamma(K)}-1}{\sqrt{\gamma(K)}+1}$ where $\gamma(K) := \max_{i,j,k,l} \frac{K_{ik}K_{jl}}{K_{jk}K_{il}}$. This result implies the exponential convergence of the Sinkhorn iterates $u^{(k)}$ and $v^{(k)}$.

## 2.4 Convergence Rate of One-Sample Estimation

Pooladian and Niles-Weed (2024) analyzed the one-sample estimation task for the Schrödinger Bridge. In this setting, full access to the source distribution $\mu$ is assumed, while the target distribution $\nu$ is observed only through the empirical distribution $\nu_n = \frac{1}{n}\sum_{j=1}^{n}\delta_{Y_j}$ based on i.i.d. samples $Y_1, \ldots, Y_n$. Let $(f_{\infty,n}^*, g_{\infty,n}^*)$ denote the optimal dual potentials solving $\mathrm{OT}_\varepsilon(\mu, \nu_n)$, and let $P_{\infty,n}^* \in \mathcal{P}(\Omega)$ be the corresponding Schrödinger Bridge. Under Assumptions 1, 2 and 3 stated below, Pooladian and Niles-Weed (2024, Theorem 4.1) proved that there exists a constant $C_\nu$, depending only on $\nu$ and geometric constants of its support $\mathrm{supp}(\nu)$, such that for any $\tau \in [0, 1)$ the following holds:

$$\mathbb{E}\left[\mathrm{TV}^2\left(P_{\infty,n}^{*,[0,\tau]}, P_{\infty,\infty}^{*,[0,\tau]}\right)\right] \le C_\nu \left\{\frac{\varepsilon^{-d_\nu-1}}{\sqrt{n}} + \frac{R^2\,\varepsilon^{-d_\nu}}{(1-\tau)^{d_\nu+2}\,n}\right\}, \tag{11}$$

where $d_\nu$ is the intrinsic dimensionality of $\mathrm{supp}(\nu)$.

## 3 Main Results

In this section, we provide statistical and algorithmic convergence rates of Sinkhorn bridge (Pooladian and Niles-Weed, 2024) in two-sample estimation setting.

---

[1]The standard Sinkhorn algorithm updates as $u^{(k+1)} = \mathbf{1}_m \oslash (mKv^{(k)})$, $v^{(k+1)} = \mathbf{1}_n \oslash (nK^T u^{(k+1)})$. This update is derived from the formulation of the primal problem 3 that incorporates the regularization term $H(\pi)$, and there is no substantive difference.

## 3.1 Sinkhorn Bridge in Two-sample Case

Given empirical distributions $\mu_m$ and $\nu_n$, the Sinkhorn bridge constructs the estimator $P_{m,n}^k \in \mathcal{P}(\Omega)$ for the Schrödinger bridge between true distributions $\mu$ and $\nu$ as follows. First, the Sinkhorn algorithm runs for $k$ iterations to solve $\mathrm{OT}_\varepsilon(\mu_m, \nu_n)$, yielding the dual potentials $f_{m,n}^{(k)}$ and $g_{m,n}^{(k)}$. Then, the estimator $P_{m,n}^k$ is defined as the path measure of the following SDE starting from $\mu$:

$$\mathrm{d}X_t = b_{m,n}^{(k)}(X_t, t)\,\mathrm{d}t + \sqrt{\varepsilon}\,\mathrm{d}B_t, \quad X_0 \sim \mu, \tag{12}$$

where the drift function $b_{m,n}^{(k)}$ is

$$b_{m,n}^{(k)}(z, t) = \frac{1}{1-t}\left(-z + \frac{\sum_{j=1}^n Y_j \exp_\varepsilon\left(g_{m,n}^{(k)}(Y_j) - \frac{1}{2(1-t)}\|Y_j - z\|_2^2\right)}{\sum_{j=1}^n \exp_\varepsilon\left(g_{m,n}^{(k)}(Y_j) - \frac{1}{2(1-t)}\|Y_j - z\|_2^2\right)}\right). \tag{13}$$

## 3.2 Convergence Rates of Sinkhorn Bridge

First, we analyze the statistical convergence rate of the path measure $P_{m,n}^* \in \mathcal{P}(\Omega)$ using the optimal Schrödinger potentials $f_{m,n}^*$ and $g_{m,n}^*$. Specifically, $P_{m,n}^*$ is defined analogously to $P_{m,n}^k$ by replacing the potential $g_{m,n}^{(k)}$ with its optimal counterpart $g_{m,n}^*$. For details of the definition, see Appendix A.1. The analysis will be carried out under the following assumptions.

**Assumption 1.** *The potentials $g_{\infty,\infty}^*, g_{\infty,n}^*, g_{m,n}^*$ satisfy*

$$\int g_{\infty,\infty}^*(y)\nu(dy) = \int g_{\infty,n}^*(y)\nu_n(dy) = \int g_{m,n}^*(y)\nu_n(dy) = 0.$$

**Assumption 2.** *The supports of $\mu$ and $\nu$ are subsets of a ball with radius $R$ centered at the origin.*

**Assumption 3.** *The measure $\nu$ is supported on a compact, smooth, connected Riemannian manifold $(N, h)$ without boundary, embedded in $\mathbb{R}^d$ and equipped with the submanifold geometry induced by its inclusion into $\mathbb{R}^d$; furthermore, $\dim N = d_\nu \geq 3$. Moreover, $\nu$ admits a Lipschitz continuous and strictly positive density with respect to the Riemannian volume measure on $N$.*

The manifold hypothesis suggests that data distributions often lie on lower-dimensional manifolds, and statistical learning methods can exploit this structure. In the EOT literature, Stromme (2023b) established the *Minimum Intrinsic Dimension* (MID) scaling, where the minimum of the intrinsic dimensions governs the convergence rate. For the Sinkhorn bridge under one-sample setting, Pooladian and Niles-Weed (2024) demonstrated that the convergence rate is independent of the ambient dimension $d$ and depends only on the intrinsic dimension of the target distribution $\nu$. Following these works, we also impose data-manifold assumptions solely on the target distribution $\nu$.

The following theorem provides the squared total variation error between $P_{\infty,\infty}^*$ and $P_{m,n}^*$.

**Theorem 1** (Statistical convergence rate). *Suppose Assumptions 1, 2 and 3 hold. Let $P_{\infty,\infty}^*$ be the path measure of the true Schrödinger bridge for marginals $\mu, \nu$. Then, it follows that for any $\tau \in [0, 1)$,*

$$\mathbb{E}[\mathrm{TV}^2(P_{\infty,\infty}^{*,[0,\tau]}, P_{m,n}^{*,[0,\tau]})]$$
$$\lesssim \frac{R^2}{n(1-\tau)^{d_\nu+2}\varepsilon^{d_\nu}} + \frac{\tau}{1-\tau}\left(\varepsilon^2 + \frac{dR^8}{\varepsilon^2}\right) \cdot \left(\frac{R}{\varepsilon}\right)^{9d_\nu+4} \cdot \left(\frac{1}{m} + \frac{1}{n}\right). \tag{14}$$

*where $P_{\infty,\infty}^{*,[0,\tau]}$ and $P_{m,n}^{*,[0,\tau]}$ are restrictions of $P_{\infty,\infty}^*$ and $P_{m,n}^*$ on the time-interval $[0, \tau]$.*

**Remark 1.** *In this section and related proofs, we suppress non-dominant terms such as polynomial factors of $d_\nu$, geometric characteristics of $\mathrm{supp}(\nu)$, and exponential terms of order $O(d_\nu)$ in uniform constants, in the notation $\lesssim$.*

We make the following observations: (1) Regarding the data dimensions, the intrinsic dimension $d_\nu$ governs the growth of the convergence rate, as in Stromme (2023b); Pooladian and Niles-Weed (2024). Although our bound, unlike theirs, involves the ambient dimension $d$, the dependence is merely linear. Moreover, we note that the term $d$ arises from the covering number of the data space

and can also be replaced by the intrinsic dimension of the source distribution $\mu$ by additionally imposing a manifold assumption on $\mu$. (2) The degeneration of the estimated SDE toward a specific sample point $Y_j$ arises as $\tau \to 1$ because the estimated drift $b_{m,n}^{(k)}$ points toward a particular sample $Y_j$ near $t = 1$, causing the deviation from the target distribution $\nu$ at the terminal time $t = 1$. Therefore, it is common practice to restrict the time interval to $[0, \tau]$ ($\tau < 1$) to ensure generalization. (3) Our result extends and sharpens the result of Pooladian and Niles-Weed (2024) to the two-sample setting, improving the convergence rate in $n$ from $n^{-1/2}$ to $n^{-1}$ at the cost of a slight deterioration regarding, $\varepsilon$, $R$, and $d$.

In the next theorem, we give the algorithmic convergence rate of $P_{m,n}^k \to P_{m,n}^*$ attained by running the Sinkhorn iterations $k \to \infty$.

**Theorem 2** (Algorithmic convergence rate). *Under Assumptions 1 and 2, we get for any $\tau \in [0, 1)$,*

$$\mathbb{E}[\text{TV}^2(P_{m,n}^{*,[0,\tau]}, P_{m,n}^{k,[0,\tau]})] \lesssim \frac{\tau}{1-\tau} \frac{R^6}{\varepsilon^2} \left( \tanh \left( \frac{R^2}{\varepsilon} \right) \right)^{4k},$$

*where $P_{m,n}^{k,[0,\tau]}$ are restrictions of $P_{m,n}^k$ on the time-interval $[0, \tau]$.*

These results immediately imply the convergence rate $\mathbb{E}[\text{TV}^2(P_{m,n}^{k,[0,\tau]}, P_{\infty,\infty}^{*,[0,\tau]})] = \mathcal{O}(1/m + 1/n + r^{4k})$ for some $r \in (0, 1)$ regarding the sample size $m$ and $n$, and the number of Sinkhorn iterations $k$.

## 3.3 Drift Estimation using Neural Networks in the Schrödinger Bridge Problem

The drift estimator (13) of the Sinkhorn bridge between $\mu_m$ and $\nu_n$ is defined as an empirical average regarding $\nu_n$. As a result, simulating the SDE (12) using a time discretization scheme (e.g., Euler–Maruyama approximation) incurs an $O(n)$-computational cost at every discretization step. To improve computational efficiency, we consider a neural network-based drift approximation.

Recall that $W_{|x_0,x_1}^\varepsilon$ is the Brownian bridge connecting $x_0$ and $x_1$, so its marginal distribution at time $t \in [0, 1]$ is a Gaussian distribution $W_{t|x_0,x_1}^\varepsilon = \mathcal{N}\big((1-t)x_0 + tx_1, \varepsilon t(1-t)I_d\big)$. For dual potentials $f_{m,n}^{(k)} \in L^1(\mu_m)$ and $g_{m,n}^{(k)} \in L^1(\nu_n)$ obtained by the Sinkhorn algorithm, we define the joint distribution $\pi_{m,n}^{(k)}$ analogously to Eq. (6) by replacing the optimal potentials and $\mu, \nu$ with $f_{m,n}^{(k)}, g_{m,n}^{(k)}$, and $\mu_m, \nu_n$. Using $\pi_{m,n}^{(k)}$, we define the mixture of Brownian bridge (a.k.a. *reciprocal process*): $\Pi^{(k)} = \int W_{|x_0,x_1}^\varepsilon \, d\pi_{m,n}^{(k)}(x_0, x_1) \in \mathcal{P}(\Omega)$. The drift function $b_{m,n}^{(k)}$ in Eq. (13) is known to have the following expression (*Markovian projection* (Shi et al., 2023)) (see Appendix B.2):

$$b_{m,n}^{(k)}(x_t, t) = \mathbb{E}_{X \sim \Pi^{(k)}} \left[ \frac{X_1 - x_t}{1 - t} \,\bigg|\, X_t = x_t \right].$$

Therefore, we see that the function $b_{m,n}^{(k)}$ minimizes the following functional $\mathcal{L}$:

$$\mathcal{L}(b) = \int_0^1 \int_{\mathbb{R}^d \times \mathbb{R}^d} \left\| b(x_t, t) - \frac{x_1 - x_t}{1 - t} \right\|_2^2 \, d\Pi_{t1}^{(k)}(x_t, x_1) \, dt,$$

where $\Pi_{t1}^{(k)}$ is the marginal distribution of $\Pi^{(k)}$ on time points $t$ and 1. This suggests training a neural network $b_\theta$ with parameter $\theta$ to minimize the above functional. We note that samples from $\Pi_{t1}^{(k)}$ can be obtained by sampling $(x_0, x_1) \sim \pi_{m,n}^{(k)}$, $t \sim U[0, \tau]$, and $x_t \sim W_{t|x_0,x_1}^\varepsilon$ and discarding $x_0$. Therefore, with this sampling scheme, SGD can be efficiently applied to minimize $\mathcal{L}(b_\theta)$. The detail of the procedure is described in Algorithm 1. This procedure yields a neural network approximation of the drift function defined by the Sinkhorn bridge.

## 3.4 Other Estimators of Schrödinger Bridge

Our theoretical results on Sinkhorn bridge (Pooladian and Niles-Weed, 2024) can be directly applied to the representative Schrödinger Bridge solvers such as [SF]$^2$M (Tong et al., 2023), lightSB(-M) (Korotin et al., 2023; Gushchin et al., 2024), DSBM-IMF (Shi et al., 2023; Peluchetti, 2023), and BM2 (Peluchetti, 2024). First, we show that the optimal estimator produced by each of these methods

---

**Algorithm 1** Drift Approximation via Neural Network

---

**input** Joint distribution $\pi_{m,n}$ defined by potentials $f \in L^1(\mu_m), g \in L^1(\nu_n)$; neural network $b_\theta$
**output** Trained neural network $b_\theta$ that approximates the drift function
   **repeat**
      Sample batch of pairs $\{x_0^i, x_1^i\}_{i=1}^N \sim \pi_{m,n}$
      Sample batch $t_i \sim U[0, \tau]$
      For each triplet $(x_0^i, x_1^i, t_i)$, sample $x_t^i \sim W_{t_i|x_0^i, x_1^i}^\varepsilon$ (Brownian bridge)
      Compute the loss:

$$L(\theta) \leftarrow \frac{1}{N} \sum_{i=1}^N \left\| b_\theta(x_t^i, t_i) - \frac{x_1^i - x_t^i}{1 - t} \right\|_2^2$$

      Update $\theta$ using $\nabla L(\theta)$
   **until** converged;

---

coincides with $P_{m,n}^*$ attained by Sinkhorn bridge with $k \to \infty$ (Proposition 1). Moreover, when DSBM-IMF and BM2 are initialized based on the reference process, the estimator after $k$ iterations matches $P_{m,n}^k$ obtained after $k$ iterations of the Sinkhorn bridge (Proposition 2). Furthermore, it follows from the proof of Proposition 1 that Algorithm 1, which learns the Sinkhorn bridge drift via a neural network, is a special case of the algorithm proposed in [SF]²M. The proofs of these results are postponed to the Appendix. Consequently, our statistical and algorithmic convergence analysis applies to these methods, endowing the theoretical guarantees.

**Proposition 1.** *The optimal estimators produced by [SF]²M, lightSB(-M), DSBM-IMF and BM2 coincide with $P_{m,n}^*$ attained by Sinkhorn bridge with $k \to \infty$.*

**Proposition 2.** *With initialization based on the reference process, both DSBM-IMF and BM2 produce the same estimator as $P_{m,n}^k$ of the Sinkhorn bridge for all iterations $k$.*

## 4 Experiments

In this section, we verify our theoretical findings through numerical experiments. In Section 4.1, we verify the dependence of the statistical error on the sample size presented in Theorem 1, and the algorithmic convergence rate presented in Theorem 2. Next, in Section 4.2, we evaluate the neural network approximation of drift coefficient as illustrated in Section 3.3. All experimental implementations are conducted with PyTorch 2.6.0 (Paszke et al., 2019).

### 4.1 Experimental Verification of Theorems 1 and 2

We use 3-dimensional normal distributions $\mu = \mathcal{N}(0, I_3)$ and $\nu = \mathcal{N}(0, B)$ as source and target distributions, where $B \in \mathbb{R}^{3 \times 3}$ is a random positive definite matrix. Under this setting, Bunne et al. (2023, Eq. (25)-(29)) provided explicit expressions for the drift coefficient $b_{\infty,\infty}^*$ and the marginal distribution $P_{\infty,\infty}^{*,t}$. We use these expressions to evaluate the errors of the drift $b_{m,n}^*$ and the corresponding Schrödinger Bridge estimators $P_{m,n}^*$ in the finite-sample settings. These estimators are approximated by running the Sinkhorn bridge with a sufficiently large number of iterations ($k \to \infty$).

**Statistical convergence.** To verify that the estimator $b_{m,n}^*$ converges to the true drift $b_{\infty,\infty}^*$ when the sample sizes $m$ and $n$ increase, we draw $m$ samples from $\mu$ and $n$ samples from $\nu$, respectively, and compute the estimator $b_{m,n}^*(\cdot, t)$ for an arbitrary time $t \in [0, 1)$. Subsequently, we evaluate the mean squared error (MSE) against the true drift $b_{\infty,\infty}^*(\cdot, t)$.

$$\text{MSE}_{\text{sample}}(m, n, t) = \mathbb{E}_{\substack{X_1,\ldots X_m \sim \mu \\ Y_1,\ldots,Y_n \sim \nu}} \|b_{m,n}^*(z, t) - b_{\infty,\infty}^*(z, t)\|_{L^2(P_{\infty,\infty}^{*,t}(z))}^2.$$

To compute this, the norm $\|\cdot\|_{L^2(P_{\infty,\infty}^{*,t})}$ is approximated using Monte Carlo with 10,000 samples drawn from $P_{\infty,\infty}^{*,t}$. The expectation over the samples is computed by averaging over 10 independent sampling trials. With a fixed parameter $\varepsilon = 0.1$, we generated heatmaps for several $t \in [0, 1)$ while varying the sample sizes $m, n$ used in the estimator definition (Figure 1).

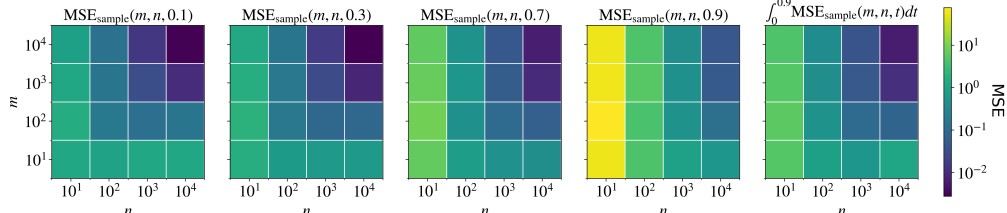

Figure 1: Heatmaps illustrating $\text{MSE}_{\text{sample}}(m, n, t)$ as a function of sample sizes $m$ and $n$ for various time points $t$. The error decreases roughly proportionally to $(m^{-1} + n^{-1})$.

As evident from the figure, the mean squared error deteriorates as $t \to 1$, but the overall convergence rate remains unchanged. For all $t$ shown in the figure, the convergence rate is observed to be approximately proportional to $(m^{-1} + n^{-1})$. This aligns with the result predicted by Theorem 1.

**Algorithmic convergence.** To verify that the estimator $P_{m,n}^{(k),[0,\tau]}$ exponentially approaches $P_{m,n}^{*,[0,\tau]}$ as the iteration count $k$ increases, we obtain $m$ and $n$ independent samples from distributions $\mu$ and $\nu$, respectively. We then consider the integral of the difference between estimators $b_{m,n}^{(k)}(\cdot, t)$ and $b_{m,n}^{*}(\cdot, t)$ over the interval $[0, \tau]$. Specifically, we evaluate the mean squared error integrated over $t \in [0, \tau]$:

$$\int_0^\tau \text{MSE}_{\text{sinkhorn}}(k, t) \, dt = \int_0^\tau \mathbb{E}_{\substack{X_1, \ldots, X_m \sim \mu \\ Y_1, \ldots, Y_n \sim \nu}} \left\| b_{m,n}^{(k)}(z, t) - b_{m,n}^{*}(z, t) \right\|_{L^2(P_{m,n}^{*,t})}^2 dt.$$

Time integrals over the interval $[0, \tau]$ are approximated via Monte Carlo integration by uniformly sampling $1,000$ time-points. The norm $\| \cdot \|_{L^2(P_{m,n}^{*,t})}$ is also estimated via Monte Carlo integration, using $1,000$ samples drawn from $P_{m,n}^{*,t}$. We set $m = n = 1,000$ and compute the expectation by averaging over 10 independent samplings. With the regularization parameter fixed at $\varepsilon = 0.005$, we generate graphs of the integral values over $[0, \tau]$ for varying $k$ and multiple $\tau$ values (Figure 2). For all $\tau$ values shown in the figure, the convergence rate exhibits exponential decay with respect to $k$. This observation corroborates the theoretical prediction in Theorem 2.

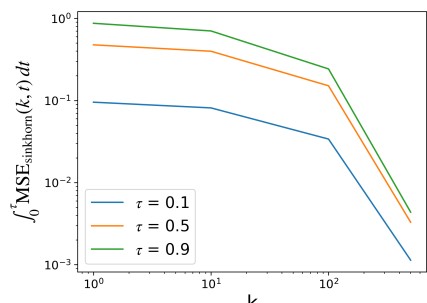

Figure 2: Integrated mean squared error as a function of Sinkhorn iterations $k$ for multiple integration intervals $[0, \tau]$.

Moreover, under the experimental setup of Section 4.2, we illustrate in Figure 3 the evolution of the drift when the number of Sinkhorn iterations is set to 1, 5, and 10. In the figure, it is observed that as the number of iterations $k$ increases, the drift $b_{m,n}^{(k)}$ rapidly converges to the optimal drift $b_{m,n}^{*}$.

### 4.2 Experimental Verification of Neural Network-Based Drift Estimation

Next, we evaluate the effectiveness of the drift approximation by a neural network using Algorithm 1. In this experiment, we set $\varepsilon = 0.1$, defined $\mu$ as the eight-Gaussians distribution and $\nu$ as the moons distribution, and drew $1,000$ independent samples from each. Using these samples, the Sinkhorn algorithm is employed to approximate the optimal EOT coupling $\pi_{m,n}^{*}$ between $\mu_m$ and $\nu_n$, and, via Algorithm 1, a neural network approximation $b_\theta$ of the drift $b_{m,n}^{*}$ is obtained. We employ an 4-layer neural network with 512-512-512 hidden neurons for $b_\theta$, and train it using the AdamW optimizer with a learning rate of $1 \times 10^{-3}$, weight decay of $1 \times 10^{-5}$, and a mini-batch size of $4,096$. Finally, starting from each sample of $\mu_m$, we simulate trajectories using either $b_{m,n}^{(k)}$ or the neural network drift $b_\theta$, and present the results in Figure 3. Trajectory simulations are performed using the Euler–Maruyama approximation with $1,000$ discretization steps.

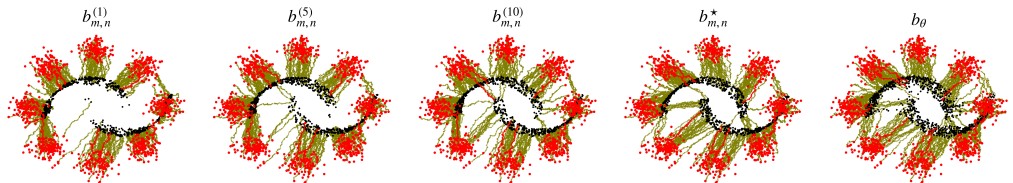

Figure 3: From left to right, the simulation results of the Schrödinger bridge using the estimated drifts $b_{m,n}^{(1)}$, $b_{m,n}^{(5)}$, and $b_{m,n}^{(10)}$ obtained by terminating the Sinkhorn iteration after 1, 5, and 10 iterations, respectively, the optimal drift $b_{m,n}^*$, and the neural network–approximated drift $b_\theta$.

## 5    Conclusion and Discussion

In this study, we provide a comprehensive analysis of statistical guarantees for the Schrödinger Bridge problem. Specifically, we establish theoretical guarantees in two key settings: (i) the two-sample estimation task and (ii) intermediate estimators during the learning process.

Our main contributions are as follows. First, we establish a statistical convergence analysis in the two-sample estimation setting for the Schrödinger bridge estimator with $k \to \infty$, which demonstrates a statistical convergence rate of $O\left(\frac{1}{n} + \frac{1}{m}\right)$. Second, we establish new convergence guarantees for the dual potentials obtained during intermediate iterations of the Sinkhorn algorithm, proving exponential convergence in the finite-sample setting. These results allow for a clear estimation of the number of samples $m, n$ and Sinkhorn iterations $k$ required to achieve a desired precision. Experimental results align with our theoretical analysis, confirming that the error decreases at a rate of approximately $(m^{-1} + n^{-1})$ with respect to the sample size, and that the error decreases exponentially with respect to the number of Sinkhorn iterations $k$. These findings strengthen the statistical guarantees for existing methods proposed in works such as Korotin et al. (2023); Peluchetti (2023); Gushchin et al. (2024); Shi et al. (2023); Peluchetti (2024) through the connections among these methods. The theoretical framework of this study deepens the understanding of generalization error in the Schrödinger Bridge problem and offers new insights into the convergence properties and stability of the Sinkhorn algorithm during intermediate iterations.

Future research directions include extending the analysis to more complex distributional settings and more general reference processes, and removing linear dependence on the ambient dimension $d$. Another interesting research direction is to explore alternative optimization methods for EOT problems and analyze their performance using our theoretical framework beyond the Sinkhorn algorithm.

## Acknowledgment

This research is supported by the National Research Foundation, Singapore, Infocomm Media Development Authority under its Trust Tech Funding Initiative, and the Ministry of Digital Development and Information under the AI Visiting Professorship Programme (award number AIVP-2024-004). Any opinions, findings and conclusions or recommendations expressed in this material are those of the author(s) and do not reflect the views of National Research Foundation, Singapore, Infocomm Media Development Authority, and the Ministry of Digital Development and Information.

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

# Appendix

## A  Omitted Proofs

### A.1  Proof of Theorem 1

We here provide the complete definitions of path measures $P_{m,n}^k$, $P_{m,n}^* \in \mathcal{P}(\Omega)$. Using the Sinkhorn iterations $(f_{m,n}^{(k)}, g_{m,n}^{(k)})$ with respect to $\mu_m$ and $\nu_n$, we define the drift function $b_{m,n}^{(k)}$ as follows:

$$
\begin{aligned}
b_{m,n}^{(k)}(z,t) &= \frac{1}{1-t}\left(-z + \frac{\sum_{j=1}^n Y_j \exp_\varepsilon\left((g_{m,n}^{(k)}(Y_j) - \frac{1}{2(1-t)}\|Y_j - z\|_2^2)\right)}{\sum_{j=1}^n \exp_\varepsilon\left((g_{m,n}^{(k)}(Y_j) - \frac{1}{2(1-t)}\|Y_j - z\|_2^2)\right)}\right) \\
&= \frac{1}{1-t}\left(-z + \frac{\sum_{j=1}^n \gamma_{m,n}^{(k),t}(Y_j, z)\, Y_j}{\sum_{j=1}^n \gamma_{m,n}^{(k),t}(Y_j, z)}\right),
\end{aligned}
$$

where

$$
\gamma_{m,n}^{(k),t}(Y_j, z) = \exp_\varepsilon\left(g_{m,n}^{(k)}(Y_j) - \frac{1}{2(1-t)}\|Y_j - z\|_2^2\right).
$$

Then, $P_{m,n}^{(k)}$ is defined as the path measure of the following SDE:

$$
\mathrm{d}X_t = b_{m,n}^{(k)}(X_t)\,\mathrm{d}t + \sqrt{\varepsilon}\,\mathrm{d}B_t, \quad X_0 \sim \mu.
$$

$P_{m,n}^* \in \mathcal{P}(\Omega)$ is also defined in the similar way by replacing the Sinkhorn iterations to the corresponding optimal Schrödinger potentials. That is, using the optimal Schrödinger potentials $(f_{m,n}^*, g_{m,n}^*)$ with respect to $\mu_m$ and $\nu_n$, we define the drift function $b_{m,n}^*$ as follows:

$$
\begin{aligned}
b_{m,n}^*(z) &= \frac{1}{1-t}\left(-z + \frac{\sum_{j=1}^n Y_j \exp_\varepsilon(g_{m,n}^*(Y_j) - \frac{1}{2(1-t)}\|Y_j - z\|_2^2)}{\sum_{j=1}^n \exp_\varepsilon(g_{m,n}^*(Y_j) - \frac{1}{2(1-t)}\|Y_j - z\|_2^2)}\right) \\
&= \frac{1}{1-t}\left(-z + \frac{\sum_{j=1}^n \gamma_{m,n}^{*,t}(Y_j, z) Y_j}{\sum_{j=1}^n \gamma_{m,n}^{*,t}(Y_j, z)}\right),
\end{aligned} \tag{15}
$$

where

$$
\gamma_{m,n}^{*,t}(Y_j, z) = \exp_\varepsilon\left(g_{m,n}^*(Y_j) - \frac{1}{2(1-t)}\|Y_j - z\|_2^2\right).
$$

Then, $P_{m,n}^*$ is defined as the path measure of the following SDE:

$$
\mathrm{d}X_t = b_{m,n}^*(X_t)\,\mathrm{d}t + \sqrt{\varepsilon}\,\mathrm{d}B_t, \quad X_0 \sim \mu. \tag{16}
$$

**Theorem 1** (Statistical convergence rate)**.** *Suppose Assumptions 1, 2 and 3 hold. Let $P_{\infty,\infty}^*$ be the path measure of the true Schrödinger bridge for marginals $\mu, \nu$. Then, it follows that for any $\tau \in [0,1)$,*

$$
\begin{aligned}
&\mathbb{E}[\mathrm{TV}^2(P_{\infty,\infty}^{*,[0,\tau]}, P_{m,n}^{*,[0,\tau]})] \\
&\lesssim \frac{R^2}{n(1-\tau)^{d_\nu+2}\varepsilon^{d_\nu}} + \frac{\tau}{1-\tau}\left(\varepsilon^2 + \frac{dR^8}{\varepsilon^2}\right) \cdot \left(\frac{R}{\varepsilon}\right)^{9d_\nu+4} \cdot \left(\frac{1}{m} + \frac{1}{n}\right).
\end{aligned} \tag{14}
$$

*where $P_{\infty,\infty}^{*,[0,\tau]}$ and $P_{m,n}^{*,[0,\tau]}$ are restrictions of $P_{\infty,\infty}^*$ and $P_{m,n}^*$ on the time-interval $[0,\tau]$.*

*Proof of Theorem 1.* The proof of Theorem 1 is based on the main idea presented in Pooladian and Niles-Weed (2024); Stromme (2023b): we introduce the following entropic plan.

$$
\mathrm{d}\bar{\pi}_{\infty,n}(x,y) := \exp_\varepsilon(\bar{f}_{\infty,n}(x) + g_{\infty,\infty}^*(y) - \frac{1}{2}\|x - y\|_2^2)\,\mathrm{d}\mu(x)\,\mathrm{d}\nu_n(y). \tag{17}
$$

Here, the function $\bar{f}_{\infty,n} : \mathbb{R}^d \to \mathbb{R}$ is defined as follows:

$$\bar{f}_{\infty,n}(x) = -\varepsilon \log \left( \frac{1}{n} \sum_{j=1}^{n} \exp_\varepsilon \left( g^*_{\infty,\infty}(Y_j) - \frac{1}{2} \|x - Y_j\|_2^2 \right) \right).$$

Denoting by $\bar{P}_{\infty,n}$ the measure obtained by substituting the coupling (17) into (8), Proposition 3.1 in Pooladian and Niles-Weed (2024) implies that the drift $\bar{b}_{\infty,n}$ with law $\bar{P}_{\infty,n}$ is expressed as follows:

$$\bar{b}_{\infty,n}(z) = \frac{1}{1-t} \left( -z + \frac{\sum_{j=1}^{n} \gamma^{*,t}_{\infty,\infty}(Y_j, z) Y_j}{\sum_{j=1}^{n} \gamma^{*,t}_{\infty,\infty}(Y_j, z)} \right).$$

By incorporating the path measure $\bar{P}_{\infty,n}$ into the bound via the triangle inequality and subsequently applying Pinsker's inequality, we obtain:

$$\mathbb{E}[\mathrm{TV}^2(P^{*,[0,\tau]}_{\infty,\infty}, P^{*,[0,\tau]}_{m,n})] \lesssim \mathbb{E}[\mathrm{TV}^2(P^{*,[0,\tau]}_{\infty,\infty}, \bar{P}^{[0,\tau]}_{\infty,n})] + \mathbb{E}[\mathrm{TV}^2(\bar{P}^{[0,\tau]}_{\infty,n}, P^{*,[0,\tau]}_{m,n})]$$
$$\lesssim \mathbb{E}[H(P^{*,[0,\tau]}_{\infty,\infty} | \bar{P}^{[0,\tau]}_{\infty,n})] + \mathbb{E}[H(\bar{P}^{[0,\tau]}_{\infty,n} | P^{*,[0,\tau]}_{m,n})].$$

We analyze these two terms separately. For the first term, we apply Proposition 4.3 of Pooladian and Niles-Weed (2024)

$$\mathbb{E}[H(P^{*,[0,\tau]}_{\infty,\infty} | \bar{P}^{[0,\tau]}_{\infty,n})] \lesssim \frac{R^2}{n(1-\tau)^{d_\nu + 2} \varepsilon^{d_\nu}}.$$

For the second term, we apply Girsanov's theorem to derive the difference between the drifts.

$$\mathbb{E}[H(\bar{P}^{[0,\tau]}_{\infty,n} | P^{*,[0,\tau]}_{m,n})]$$
$$\leq \int_0^\tau \mathbb{E} \left\| \bar{b}_{\infty,n} - b^*_{m,n} \right\|^2_{L^2(\bar{P}^t_{\infty,n})} \mathrm{d}t$$
$$= \int_0^\tau \frac{1}{(1-t)^2} \mathbb{E} \left\| \frac{\sum_{j=1}^{n} \gamma^{*,t}_{\infty,\infty}(Y_j, z) Y_j}{\sum_{j=1}^{n} \gamma^{*,t}_{\infty,\infty}(Y_j, z)} - \frac{\sum_{j=1}^{n} \gamma^{*,t}_{m,n}(Y_j, z) Y_j}{\sum_{j=1}^{n} \gamma^{*,t}_{m,n}(Y_j, z)} \right\|^2_{L^2(\bar{P}^t_{\infty,n})} \mathrm{d}t$$
$$\lesssim \int_0^\tau \frac{1}{(1-t)^2} \left( \varepsilon^2 + \frac{dR^8}{\varepsilon^2} \right) \cdot \left( \frac{R}{\varepsilon} \right)^{9d_\nu + 4} \cdot \left( \frac{1}{m} + \frac{1}{n} \right) \mathrm{d}t$$
$$= \frac{\tau}{1-\tau} \left( \varepsilon^2 + \frac{dR^8}{\varepsilon^2} \right) \cdot \left( \frac{R}{\varepsilon} \right)^{9d_\nu + 4} \cdot \left( \frac{1}{m} + \frac{1}{n} \right),$$

where for the second inequality, we used Lemma 1. $\qquad\square$

**Lemma 1.** *Under assumptions of Theorem 1, we have*

$$\mathbb{E} \left\| \frac{\sum_{j=1}^{n} \gamma^{*,t}_{\infty,\infty}(Y_j, \cdot) Y_j}{\sum_{j=1}^{n} \gamma^{*,t}_{\infty,\infty}(Y_j, \cdot)} - \frac{\sum_{j=1}^{n} \gamma^{*,t}_{m,n}(Y_j, \cdot) Y_j}{\sum_{j=1}^{n} \gamma^{*,t}_{m,n}(Y_j, \cdot)} \right\|^2_{L^2(p^t_{\infty,\infty})}$$
$$\lesssim \left( \varepsilon^2 + \frac{dR^8}{\varepsilon^2} \right) \cdot \left( \frac{R}{\varepsilon} \right)^{9d_\nu + 4} \cdot \left( \frac{1}{m} + \frac{1}{n} \right).$$

*Proof of Lemma 1.* We perform the expansion while keeping $Y_j$ and $z$ fixed

$$\left\| \frac{\sum_{j=1}^{n} \gamma^{*,t}_{\infty,\infty}(Y_j, z) Y_j}{\sum_{j=1}^{n} \gamma^{*,t}_{\infty,\infty}(Y_j, z)} - \frac{\sum_{j=1}^{n} \gamma^{*,t}_{m,n}(Y_j, z) Y_j}{\sum_{j=1}^{n} \gamma^{*,t}_{m,n}(Y_j, z)} \right\|_2$$
$$= \left\| \sum_{i=1}^{n} \left( \frac{\gamma^{*,t}_{\infty,\infty}(Y_j, z)}{\sum_{j=1}^{n} \gamma^{*,t}_{\infty,\infty}(Y_j, z)} - \frac{\gamma^{*,t}_{m,n}(Y_j, z)}{\sum_{j=1}^{n} \gamma^{*,t}_{m,n}(Y_j, z)} \right) Y_j \right\|_2$$
$$\leq R \sum_{i=1}^{n} \left| \frac{\gamma^{*,t}_{\infty,\infty}(Y_j, z)}{\sum_{j=1}^{n} \gamma^{*,t}_{\infty,\infty}(Y_j, z)} - \frac{\gamma^{*,t}_{m,n}(Y_j, z)}{\sum_{j=1}^{n} \gamma^{*,t}_{m,n}(Y_j, z)} \right|.$$

From the case $(p, q) = (1, \infty)$ of Lemma 12, we have

$$
\left\| \frac{\sum_{j=1}^{n} \gamma_{\infty,\infty}^{*,t}(Y_j, z) Y_j}{\sum_{j=1}^{n} \gamma_{\infty,\infty}^{*,t}(Y_j, z)} - \frac{\sum_{j=1}^{n} \gamma_{m,n}^{*,t}(Y_j, z) Y_j}{\sum_{j=1}^{n} \gamma_{m,n}^{*,t}(Y_j, z)} \right\|_2
$$

$$
\leq \frac{R}{\varepsilon} \max_{j \in [n]} \left| \left( g_{\infty,\infty}^{*}(Y_j) - \frac{1}{2(1-t)} \|Y_j - z\|_2^2 \right) - \left( g_{m,n}^{*}(Y_j) - \frac{1}{2(1-t)} \|Y_j - z\|_2^2 \right) \right|
$$

$$
\leq \frac{R}{\varepsilon} \left\| g_{\infty,\infty}^{*} - g_{m,n}^{*} \right\|_{L^\infty(\nu)}.
$$

By applying Lemma 11, we obtain

$$
\mathbb{E} \left\| \frac{\sum_{j=1}^{n} \gamma_{\infty,\infty}^{*,t}(Y_j, z) Y_j}{\sum_{j=1}^{n} \gamma_{\infty,\infty}^{*,t}(Y_j, z)} - \frac{\sum_{j=1}^{n} \gamma_{m,n}^{*,t}(Y_j, z) Y_j}{\sum_{j=1}^{n} \gamma_{m,n}^{*,t}(Y_j, z)} \right\|_{L^2(\bar{P}_{\infty,n}^{t})}^2
$$

$$
\lesssim \frac{R^2}{\varepsilon^2} \mathbb{E} \left\| g_{m,n}^{*} - g_{\infty,\infty}^{*} \right\|_{L^\infty(\nu)}^2
$$

$$
\lesssim \left( \varepsilon^2 + \frac{dR^8}{\varepsilon^2} \right) \cdot \left( \frac{R}{\varepsilon} \right)^{9 d_\nu + 4} \cdot \left( \frac{1}{m} + \frac{1}{n} \right).
$$

$\square$

### A.2 Proof of Theorem 2

**Theorem 2** (Algorithmic convergence rate). *Under Assumptions 1 and 2, we get for any $\tau \in [0, 1)$,*

$$
\mathbb{E}[\mathrm{TV}^2(P_{m,n}^{*,[0,\tau]}, P_{m,n}^{k,[0,\tau]})] \lesssim \frac{\tau}{1 - \tau} \frac{R^6}{\varepsilon^2} \left( \tanh \left( \frac{R^2}{\varepsilon} \right) \right)^{4k},
$$

*where $P_{m,n}^{k,[0,\tau]}$ are restrictions of $P_{m,n}^{k}$ on the time-interval $[0, \tau]$.*

*Proof of Theorem 2.* We start by applying Girsanov's theorem to obtain a difference in the drifts

$$
\mathbb{E}[H(P_{m,n}^{*,[0,\tau]} | P_{m,n}^{k,[0,\tau]})]
$$

$$
\leq \int_0^\tau \mathbb{E} \left\| b_{m,n}^{*} - b_{m,n}^{k} \right\|_{L^2(P_{m,n}^{*,t})}^2 \mathrm{d}t
$$

$$
= \int_0^\tau \mathbb{E} \left\| \frac{1}{1-t} \left( \frac{\sum_{j=1}^{n} \gamma_{m,n}^{*,t}(Y_j, \cdot) Y_j}{\sum_{j=1}^{n} \gamma_{m,n}^{*,t}(Y_j, \cdot)} - \frac{\sum_{j=1}^{n} \gamma_{m,n}^{k,t}(Y_j, \cdot) Y_j}{\sum_{j=1}^{n} \gamma_{m,n}^{k,t}(Y_j, \cdot)} \right) \right\|_{L^2(P_{m,n}^{*,t})}^2 \mathrm{d}t
$$

$$
\lesssim \int_0^\tau \frac{1}{(1-t)^2} \frac{R^6}{\varepsilon^2} \left( \tanh \left( \frac{R^2}{\varepsilon} \right) \right)^{4k} \mathrm{d}t
$$

$$
= \frac{\tau}{1 - \tau} \frac{R^6}{\varepsilon^2} \left( \tanh \left( \frac{R^2}{\varepsilon} \right) \right)^{4k},
$$

where we applied Lemma 2. $\square$

**Lemma 2.** *Under assumptions of Theorem 1, we have*

$$
\mathbb{E} \left\| \frac{\sum_{j=1}^{n} \gamma_{m,n}^{*,t}(Y_j, z) Y_j}{\sum_{j=1}^{n} \gamma_{m,n}^{*,t}(Y_j, z)} - \frac{\sum_{j=1}^{n} \gamma_{m,n}^{k,t}(Y_j, z) Y_j}{\sum_{j=1}^{n} \gamma_{m,n}^{k,t}(Y_j, z)} \right\|_{L^2(P_{m,n}^{*,t})}^2 \lesssim \frac{R^6}{\varepsilon^2} \left( \tanh \left( \frac{R^2}{\varepsilon} \right) \right)^{4k}.
$$

*Proof of Lemma 2.* We perform the expansion while keeping $Y_j$ and $z$ fixed

$$
\left\| \frac{\sum_{j=1}^{n} \gamma_{m,n}^{*,t}(Y_j, z) Y_j}{\sum_{j=1}^{n} \gamma_{m,n}^{*,t}(Y_j, z)} - \frac{\sum_{j=1}^{n} \gamma_{m,n}^{k,t}(Y_j, z) Y_j}{\sum_{j=1}^{n} \gamma_{m,n}^{k,t}(Y_j, z)} \right\|_2
$$

$$= \left\| \sum_{j=1}^{n} \left( \frac{\gamma_{m,n}^{*,t}(Y_j, z)}{\sum_{j=1}^{n} \gamma_{m,n}^{*,t}(Y_j, z)} - \frac{\gamma_{m,n}^{k,t}(Y_j, z)}{\sum_{j=1}^{n} \gamma_{m,n}^{k,t}(Y_j, z)} \right) Y_j \right\|_2$$

$$\leq \sum_{j=1}^{n} \left| \frac{\gamma_{m,n}^{*,t}(Y_j, z)}{\sum_{j=1}^{n} \gamma_{m,n}^{*,t}(Y_j, z)} - \frac{\gamma_{m,n}^{k,t}(Y_j, z)}{\sum_{j=1}^{n} \gamma_{m,n}^{k,t}(Y_j, z)} \right| \|Y_j\|_2$$

$$\leq R \sum_{j=1}^{n} \left| \frac{\gamma_{m,n}^{*,t}(Y_j, z)}{\sum_{j=1}^{n} \gamma_{m,n}^{*,t}(Y_j, z)} - \frac{\gamma_{m,n}^{k,t}(Y_j, z)}{\sum_{j=1}^{n} \gamma_{m,n}^{k,t}(Y_j, z)} \right|.$$

We define $\mathbf{w}^*, \mathbf{w}^k, \mathbf{v}^*, \mathbf{v}^k \in \mathbb{R}^n$ as follows:

$$\mathbf{w}_j^* := \frac{\gamma_{m,n}^{*,t}(Y_j, z)}{\sum_{j=1}^{n} \gamma_{m,n}^{*,t}(Y_j, z)}, \quad \mathbf{w}_j^k := \frac{\gamma_{m,n}^{k,t}(Y_j, z)}{\sum_{j=1}^{n} \gamma_{m,n}^{k,t}(Y_j, z)},$$

$$\mathbf{v}_j^* := \exp\left( \frac{g_{m,n}^*(Y_j)}{\varepsilon} \right), \quad \mathbf{v}_j^k := \exp\left( \frac{g_{m,n}^{(k)}(Y_j)}{\varepsilon} \right).$$

where both $\mathbf{w}^*$ and $\mathbf{w}^k$ satisfy the conditions of a probability vector, i.e., $\mathbf{w}_j^*, \mathbf{w}_j^k > 0, \sum_{j=1}^{n} \mathbf{w}_j^* = \sum_{j=1}^{n} \mathbf{w}_j^k = 1$.

$$\left\| \frac{\sum_{j=1}^{n} \gamma_{m,n}^{*,t}(Y_j, z) Y_j}{\sum_{j=1}^{n} \gamma_{m,n}^{*,t}(Y_j, z)} - \frac{\sum_{j=1}^{n} \gamma_{m,n}^{k,t}(Y_j, z) Y_j}{\sum_{j=1}^{n} \gamma_{m,n}^{k,t}(Y_j, z)} \right\|_2^2$$

$$\leq \left( R \sum_{j=1}^{n} \left| \frac{\gamma_{m,n}^{*,t}(Y_j, z)}{\sum_{j=1}^{n} \gamma_{m,n}^{*,t}(Y_j, z)} - \frac{\gamma_{m,n}^{k,t}(Y_j, z)}{\sum_{j=1}^{n} \gamma_{m,n}^{k,t}(Y_j, z)} \right| \right)^2$$

$$= R^2 \|\mathbf{w}^* - \mathbf{w}^k\|_1^2$$

$$\lesssim R^2 d_{\text{Hilb}}(\mathbf{w}^*, \mathbf{w}^k)^2 \qquad \text{(using Lemma 13)}$$

$$= R^2 d_{\text{Hilb}}(\mathbf{v}^*, \mathbf{v}^k)^2 \qquad \text{(using Lemma 14)}$$

$$\leq R^2 \lambda(K)^{4k} d_{\text{Hilb}}(\mathbf{v}^*, \mathbf{v}^0)^2$$

$$= R^2 \lambda(K)^{4k} d_{\text{Hilb}}(\mathbf{v}^*, \mathbb{1}_m)^2$$

$$= R^2 \lambda(K)^{4k} \left( \log\left( \left( \max_j \exp_\varepsilon \left( g_{m,n}^*(Y_j) \right) \right) \left( \max_j \exp_\varepsilon \left( -g_{m,n}^*(Y_j) \right) \right) \right) \right)^2$$

$$\leq R^2 \lambda(K)^{4k} \left( \log \left( \exp_\varepsilon \left( 4R^2 \right) \exp_\varepsilon \left( 4R^2 \right) \right) \right)^2 \qquad \text{(using Lemma 3)}$$

$$\lesssim \frac{R^6}{\varepsilon^2} \lambda(K)^{4k}.$$

Since

$$\gamma(K) = \max_{ijkl} \frac{K_{ik} K_{jl}}{K_{jk} K_{il}} = \max_{ijkl} \exp\left( \frac{1}{\varepsilon} (X_i - X_j)^T (Y_k - Y_l) \right) \leq \exp_\varepsilon \left( 4R^2 \right),$$

the quantity $\lambda(K)$ can be upper bounded as follows:

$$\lambda(K) = \frac{\sqrt{\gamma(K)} - 1}{\sqrt{\gamma(K)} + 1}$$

$$\leq \frac{\exp_\varepsilon \left( 2R^2 \right) - 1}{\exp_\varepsilon \left( 2R^2 \right) + 1} \qquad \left( \because \frac{x-1}{x+1} \; (x > 0) \text{ is increasing function} \right)$$

$$= \tanh\left( \frac{R^2}{\varepsilon} \right).$$

Since the upper bound of $\lambda(K)$ is independent of $Y_j$ and $z$, the proof is complete. $\qquad \square$

## A.3 Proof of Proposition 1

**Proposition 1.** *The optimal estimators produced by [SF]²M, lightSB(-M), DSBM-IMF and BM2 coincide with $P_{m,n}^*$ attained by Sinkhorn bridge with $k \to \infty$.*

*Proof of Proposition 1.* It suffices to show that the drift estimated by each method coincides with $b_{m,n}^*$.

### Proof of optimal estimator for [SF]²M

The SDE representation of the Brownian bridge $W_{|x_0,x_1}^\varepsilon$ with endpoints $(x_0, x_1)$ is given by

$$\mathrm{d}X_t = \frac{x_1 - X_t}{1 - t}\, \mathrm{d}t + \sqrt{\varepsilon}\, \mathrm{d}B_t, \quad X_0 = x_0. \tag{18}$$

The marginal density of the Brownian bridge at time $t$ is $p_t(x_t|x_0,x_1) = \mathcal{N}(x_t|(1-t)x_0 + tx_1, \varepsilon t(1-t)I_d)$. There exists an ordinary differential equation (ODE) that preserves the same marginal distributions $p_t(x_t|x_0,x_1)$ as the SDE (18). This ODE, referred to as the *probability flow ODE*, takes the form

$$
\begin{aligned}
\mathrm{d}X_t &= \left( \frac{x_1 - X_t}{1-t} - \frac{\varepsilon}{2}\, \nabla \log p_t(X_t|x_0,x_1) \right) \mathrm{d}t \\
&= \left( \frac{1 - 2t}{2\, t(1-t)}\, X_t + \frac{1}{2(1-t)}\, x_1 - \frac{1}{2t}\, x_0 \right) \mathrm{d}t \\
&=: u_t^\circ(X_t|x_0,x_1)\, \mathrm{d}t.
\end{aligned}
$$

Consequently, the drift term of the Brownian bridge SDE can be decomposed into the sum of the probability flow ODE drift $u_t^\circ(x_t|x_0,x_1)$ and the score function $\nabla \log p_t(x_t|x_0,x_1)$:

$$\frac{x_1 - x_t}{1 - t} = u_t^\circ(x_t|x_0,x_1) + \frac{\varepsilon}{2}\, \nabla \log p_t(x_t|x_0,x_1).$$

In the [SF]²M framework, let $\pi_{m,n}^*$ denote the optimal coupling for the entropic optimal transport (EOT) between $\mu_m$ and $\nu_n$, and consider the mixture distribution $\Pi^* = \pi_{m,n}^* W_{|x_0,x_1}^\varepsilon$. Under $\Pi^*$, two neural networks $v_\theta$ and $s_\varphi$ are trained to minimize the following objectives:

$$v_{\theta^*} = \arg\min_{v_\theta} \mathbb{E}_{X \sim \Pi^*} \left[ \int_0^1 \| v_\theta(X_t, t) - u_t^\circ(X_t|X_0, X_1) \|_2^2\, \mathrm{d}t \right],$$

$$s_{\varphi^*} = \arg\min_{s_\varphi} \mathbb{E}_{X \sim \Pi^*} \left[ \int_0^1 \| s_\varphi(X_t, t) - \nabla \log p_t(X_t|X_0, X_1) \|_2^2\, \mathrm{d}t \right].$$

The estimator for the Schrödinger bridge drift in [SF]²M is then defined as

$$v_{\theta^*}(x_t, t) + \frac{\varepsilon}{2}\, s_{\varphi^*}(x_t, t).$$

When both models are sufficiently expressive, the optimal solutions admit the following conditional expectation representations:

$$v_{\theta^*}(x_t, t) = \mathbb{E}_{X \sim \Pi^*} \left[ u_t^\circ(x_t|X_0, X_1) \mid X_t = x_t \right],$$

$$s_{\varphi^*}(x_t, t) = \mathbb{E}_{X \sim \Pi^*} \left[ \nabla \log p_t(x_t|X_0, X_1) \mid X_t = x_t \right].$$

Therefore, the drift estimator in [SF]²M can be written as

$$
\begin{aligned}
&v_{\theta^*}(x_t, t) + \frac{\varepsilon}{2}\, s_{\varphi^*}(x_t, t) \\
&= \mathbb{E}_{X \sim \Pi^*} \left[ u_t^\circ(x_t|X_0, X_1) | X_t = x_t \right] + \frac{\varepsilon}{2} \mathbb{E}_{X \sim \Pi^*} \left[ \nabla \log p_t(x_t|X_0, X_1) | X_t = x_t \right] \\
&= \mathbb{E}_{X \sim \Pi^*} \left[ u_t^\circ(x_t|X_0, X_1) + \frac{\varepsilon}{2} \nabla \log p_t(x_t|X_0, X_1) | X_t = x_t \right] \\
&= \mathbb{E}_{X \sim \Pi^*} \left[ \frac{X_1 - x_t}{1 - t} \,\Big|\, X_t = x_t \right]
\end{aligned}
$$

$$= b_{m,n}^*(x_t, t).$$

**Proof of optimal estimator for LightSB(-M)**

Let $\mathcal{S}(\mu)$ denote the set of Schrödinger bridges whose source distribution is $\mu$. In lightSB(-M), the drift estimator $g_v$ is defined with a function $v : \mathbb{R}^d \to \mathbb{R}$ as follows:

$$g_v(x_t, t) = \varepsilon \nabla_{x_t} \log \left( \int \mathcal{N}\left(x' \mid x_t, \varepsilon(1-t)I_d\right) \exp\left(\frac{\|x'\|_2^2}{2}\right) v(x')\, dx' \right),$$

and let $S_v$ be the law of the SDE

$$\mathrm{d}X_t = g_v(X_t, t)\, \mathrm{d}t + \sqrt{\varepsilon}\, \mathrm{d}B_t, \qquad X_0 \sim \mu_m.$$

For any coupling $\pi_{m,n} \in \Pi(\mu_m, \nu_n)$, consider its Brownian-bridge mixture $\Pi = \pi_{m,n} W_{|x_0,x_1}^\varepsilon$. Then, by (Gushchin et al., 2024, Theorem 3.1),

$$\operatorname*{argmin}_{S \in \mathcal{S}(\mu_m)} H(\Pi \mid S) = \Pi^*.$$

Restricting $S$ to $S_v$, one obtains (Gushchin et al., 2024, Theorem 3.2)

$$H(\Pi \mid S_v) = C(\pi_{m,n}) + \frac{1}{2\varepsilon} \int_0^1 \int \left\| g_v(X_t, t) - \frac{X_1 - X_t}{1 - t} \right\|_2^2 \mathrm{d}\Pi_{t1}(X_t, X_1)\, \mathrm{d}t,$$

where $\Pi_{t1}$ denotes the marginal law of $\Pi$ at times $t$ and $1$, and $C(\pi_{m,n})$ is a constant independent of $S_v$. lightSB(-M) trains a parameterised model $v_\theta$ that minimises

$$v_{\theta*} = \operatorname*{argmin}_{v_\theta} H(\Pi \mid S_{v_\theta}) = \operatorname*{argmin}_{v_\theta} \int_0^1 \int \left\| g_{v_\theta}(X_t, t) - \frac{X_1 - X_t}{1 - t} \right\|_2^2 \mathrm{d}\Pi_{t1}(X_t, X_1)\, \mathrm{d}t.$$

The drift $g_{v_{\theta*}}$ obtained from the learned $v_{\theta*}$ serves as the drift estimator produced by lightSB(-M).

When the model capacity is sufficiently large, $g_{v_{\theta*}}$ satisfies

$$g_{v_{\theta*}}(x_t, t) = \mathbb{E}_{X \sim \Pi^*} \left[ \frac{X_1 - x_t}{1 - t} \,\middle|\, X_t = x_t \right] = b_{m,n}^*(x_t, t).$$

**Proof of optimal estimator for DSBM-IMF and BM2**

Let $F(v_f)$ and $B(v_b)$ denote the laws induced by the following SDEs:

$$\mathrm{d}X_t = v_f(X_t, t)\, \mathrm{d}t + \sqrt{\varepsilon}\, \mathrm{d}B_t, \quad X_0 \sim \mu_m,$$
$$\mathrm{d}X_t = v_b(X_t, t) dt + \sqrt{\varepsilon}\, \mathrm{d}\bar{B}_t, \quad X_1 \sim \nu_n.$$

Here, in the second SDE, $dt$ represents an infinitesimal negative time step, and $\bar{B}_t$ denotes the time-reversed Brownian motion.

Given drifts $v_{f'}$ and $v_{b'}$, define the following loss functions:

$$L_{\text{forward}}(v_f; v_{b'}) := \mathbb{E}_{X \sim B(v_{b'})_{01} W_{|x_0,x_1}^\varepsilon} \left[ \int_0^1 \left\| v_f(X_t, t) - \frac{X_1 - X_t}{1 - t} \right\|_2^2 \mathrm{d}t \right],$$

$$L_{\text{backward}}(v_b; v_{f'}) := \mathbb{E}_{X \sim F(v_{f'})_{01} W_{|x_0,x_1}^\varepsilon} \left[ \int_0^1 \left\| v_b(X_t, t) - \frac{X_t - X_0}{t} \right\|_2^2 \mathrm{d}t \right].$$

In DSBM-IMF, starting from initial drifts $v_f^{(0)}$ and $v_b^{(0)}$, we perform the following iterative updates:

$$v_f^{(k+1)} = \operatorname*{argmin}_{v_f} L_{\text{forward}}(v_f; v_b^{(k)}), \quad v_b^{(k+1)} = \operatorname*{argmin}_{v_b} L_{\text{backward}}(v_b; v_f^{(k+1)}).$$

In BM2, given initial drifts $v_f^{(0)}$ and $v_b^{(0)}$, the iterative procedure is:

$$v_f^{(k+1)} = \operatorname*{argmin}_{v_f} L_{\text{forward}}(v_f; v_b^{(k)}), \quad v_b^{(k+1)} = \operatorname*{argmin}_{v_b} L_{\text{backward}}(v_b; v_f^{(k)}).$$

For both methods, $F(v_f^{(k)})$ and $B(v_b^{(k)})$ converge to $\Pi^*$ as $k \to \infty$ (Shi et al., 2023, Theorem 8), (Peluchetti, 2024, Lemma 1).

Therefore, letting $v_f^*$ and $v_b^*$ denote the limits of $v_f^{(k)}$ and $v_b^{(k)}$ as $k \to \infty$, we have that $v_f^*$ equals $b_{m,n}^*$:

$$
\begin{aligned}
v_f^*(x_t, t) &= \mathbb{E}_{X \sim B(v_b^*)_{01} W_{|x_0,x_1}^\varepsilon} \left[ \frac{X_1 - x_t}{1 - t} \,\middle|\, X_t = x_t \right] \\
&= \mathbb{E}_{X \sim \Pi^*} \left[ \frac{X_1 - x_t}{1 - t} \,\middle|\, X_t = x_t \right] \\
&= b_{m,n}^*(x_t, t).
\end{aligned}
$$

$\square$

## A.4 Proof of Proposition 2

**Proposition 2.** *With initialization based on the reference process, both DSBM-IMF and BM2 produce the same estimator as $P_{m,n}^k$ of the Sinkhorn bridge for all iterations $k$.*

*Proof of Proposition 2.* We begin by introducing the Iterative Proportional Fitting (IPF) method (Fortet, 1940; Kullback, 1968; Rüschendorf, 1995). IPF provides one means of solving Eq. (7) and generates a sequence of path measures $(\tilde{P}^{(k)})_{k \in \mathbb{N}}$ according to

$$
\tilde{P}^{(2k+1)} = \underset{\tilde{P}}{\arg\min} \left\{ H(\tilde{P} \,|\, \tilde{P}^{(2k)}) \,\middle|\, \tilde{P}_0 = \mu_m \right\},
$$

$$
\tilde{P}^{(2k+2)} = \underset{\tilde{P}}{\arg\min} \left\{ H(\tilde{P} \,|\, \tilde{P}^{(2k+1)}) \,\middle|\, \tilde{P}_1 = \nu_n \right\}.
$$

We initialize with $\tilde{P}^{(0)} = W^\varepsilon$. It is known (Léonard, 2013; Nutz, 2021) that the sequence $(\tilde{P}^{(k)})_{k \in \mathbb{N}}$ satisfies

$$
\tilde{P}^{(2k+1)} = \pi^{f_{m,n}^{(k+1)}, g_{m,n}^{(k)}} W_{|x_0,x_1}^\varepsilon,
$$

$$
\tilde{P}^{(2k+2)} = \pi^{f_{m,n}^{(k+1)}, g_{m,n}^{(k+1)}} W_{|x_0,x_1}^\varepsilon = \Pi^{(k+1)}.
$$

Here the coupling $\pi^{f,g}$ is defined by

$$
\mathrm{d}\pi^{f,g}(x, y) = \exp_\varepsilon \left( f(x) + g(y) - \tfrac{1}{2}\|x - y\|_2^2 \right) \mathrm{d}\mu_m(x)\, \mathrm{d}\nu_n(y).
$$

Hence it suffices to show that the drifts estimated by each algorithm coincide with $b_{m,n}^{(k)}$.

When the initial drift for DSBM is set to $v_b^{(0)} = 0$, the sequence $B(v_b^{(0)}), F(v_f^{(1)}), B(v_b^{(1)}), F(v_f^{(2)}), B(v_b^{(3)}), \ldots$ coincides with the IPF sequence $(\tilde{P}^{(k)})_{k \in \mathbb{N}}$ (Shi et al., 2023, Proposition 10). Consequently, $v_f^{(k+1)}$ coincides with $b_{m,n}^{(k)}$.

$$
\begin{aligned}
v_f^{(k+1)}(x_t, t) &= \mathbb{E}_{X \sim B(v_b^{(k)})_{01} W_{|x_0,x_1}^\varepsilon} \left[ \frac{X_1 - x_t}{1 - t} \,\middle|\, X_t = x_t \right] \\
&= \mathbb{E}_{X \sim \pi^{f_{m,n}^{(k)}, g_{m,n}^{(k)}} W_{|x_0,x_1}^\varepsilon} \left[ \frac{X_1 - x_t}{1 - t} \,\middle|\, X_t = x_t \right] \\
&= \mathbb{E}_{X \sim \Pi^{(k)}} \left[ \frac{X_1 - x_t}{1 - t} \,\middle|\, X_t = x_t \right] \\
&= b_{m,n}^{(k)}(x_t, t).
\end{aligned}
$$

When the initial drift for BM2 is initialized as $v_f^{(0)} = v_b^{(0)} = 0$, the sequence $B(v_b^{(0)}), F(v_f^{(1)}), B(v_b^{(2)}), \ldots$ coincides with the IPF sequence $(\tilde{P}^{(k)})_{k \in \mathbb{N}}$ (Peluchetti, 2024, Theorem 1). Thus, $v_f^{(2k+1)}$ is given by

$$
v_f^{(2k+1)}(x_t, t) = \mathbb{E}_{X \sim B(v_b^{(2k)})_{01} W_{|x_0,x_1}^\varepsilon} \left[ \frac{X_1 - x_t}{1 - t} \,\middle|\, X_t = x_t \right]
$$

$$= \mathbb{E}_{X \sim \pi^{f_{m,n}^{(k)}, g_{m,n}^{(k)}} W_{|x_0, x_1}^{\varepsilon}} \left[ \frac{X_1 - x_t}{1 - t} \,\middle|\, X_t = x_t \right]$$

$$= \mathbb{E}_{X \sim \Pi^{(k)}} \left[ \frac{X_1 - x_t}{1 - t} \,\middle|\, X_t = x_t \right]$$

$$= b_{m,n}^{(k)}(x_t, t).$$

Similarly, the sequence $F(v_f^{(0)}), B(v_b^{(1)}), F(v_f^{(2)}), \ldots$ corresponds to IPF with the update order reversed, and the same conclusion holds for $v_f^{(2k)}$.

$\square$

## B Technical Lemmas

### B.1 Lemmas used for Theorems 1 and 2

**Lemma 3** (Stromme, 2023b, Proposition 14). *Under assumptions of Theorem 1, we have*
$$\|f_{\infty,\infty}^*\|_\infty, \|g_{\infty,\infty}^*\|_\infty \le 2R^2, \quad \|f_{m,n}^*\|_\infty, \|g_{m,n}^*\|_\infty \le 4R^2.$$

In the following, $p_{\infty,\infty}^*$ denotes the Radon-Nikodym derivative regarding $\mu \otimes \nu$, defined in Eq. (6), and $p_{m,n}^*$ is also defined in a similar manner replacing $f_{\infty,\infty}^*, g_{\infty,\infty}^*$ with $f_{m,n}^*, g_{m,n}^*$.

**Lemma 4** (Stromme, 2023b, Proposition 15). *The population dual potentials $f_{\infty,\infty}^*$ and $g_{\infty,\infty}^*$ are $2R$-Lipschitz over $\mathrm{supp}(\mu)$ and $\mathrm{supp}(\nu)$, respectively. The extended empirical dual potentials $f_{m,n}^*$ and $g_{m,n}^*$ are also $2R$-Lipschitz over $\mathrm{supp}(\mu)$ and $\mathrm{supp}(\nu)$, respectively. In particular, the population density $p_{\infty,\infty}^*$ and the extended empirical density $p_{m,n}^*$ are each $\frac{4R}{\varepsilon}$-log-Lipschitz in each of their variables over $\mathrm{supp}(\mu \otimes \nu)$.*

**Lemma 5** (Stromme, 2023b, Lemma 25). *Under assumptions of Theorem 1, with probability at least $1 - \frac{1}{n}e^{-20R^2/\varepsilon}$*

$$\inf_{z \in \mathrm{supp}(\nu)} \nu_n \left( B\left( z, \frac{\varepsilon}{2R} \right) \right) \gtrsim \left( \frac{\varepsilon}{R} \right)^{d_\nu}. \tag{19}$$

**Lemma 6** (Stromme, 2023b, Lemma 26). *Under assumptions of Theorem 1, we have*

$$\|p_{\infty,\infty}^*\|_{L^\infty(\mu \otimes \nu)} \lesssim \left( \frac{R}{\varepsilon} \right)^{d_\nu}. \tag{20}$$

*And, with probability at least $1 - \frac{1}{n}e^{-20R^2/\varepsilon}$*

$$\|p_{m,n}^*\|_{L^\infty(\mu \otimes \nu)} \lesssim \left( \frac{R}{\varepsilon} \right)^{d_\nu}. \tag{21}$$

**Lemma 7** (Stromme, 2023b, Lemma 27 adapted). *Under the assumptions of Theorem 1, if $\varepsilon/R$ is sufficiently small, then*

$$\mathbb{E}\|g_{m,n}^* - g_{\infty,\infty}^*\|_{L^2(\nu_n)}^2 \lesssim \left( \varepsilon^2 + \frac{R^8}{\varepsilon^2} \right) \cdot \left( \frac{R}{\varepsilon} \right)^{6d_\nu + 4} \cdot \left( \frac{1}{m} + \frac{1}{n} \right). \tag{22}$$

Lemma 7 are the extensions of Stromme (2023b, Lemma 27) that only treats the case of $m = n$. This extension can be readily verified by a slight modification of the proof in Stromme (2023b).

**Lemma 8.** *Let $U_m(y) = (\mu_m - \mu)\big(p_{\infty,\infty}^*(\cdot, y)\big)$. Then the following holds:*

$$\mathbb{E} \sup_{y \in \mathrm{supp}(\nu)} U_m(y)^2 \lesssim \frac{R^2}{m} \left( \frac{R}{\varepsilon} \right)^{2d_\nu + 2}.$$

*Proof.* We first note that $\mu(p_{\infty,\infty}^*(\cdot, y)) = 1$ since the marginal of $\pi_{\infty,\infty}^*$ on $y$ is $\nu$, and hence $U_m(y) = \mu_m(p_{\infty,\infty}^*(\cdot, y)) - 1$. Hoeffding's lemma, with $0 \le p_{\infty,\infty}^* \lesssim \left( \frac{R}{\varepsilon} \right)^{d_\nu}$, implies that there exists a constant $C > 0$ such that

$$\mathbb{E} \exp\{s U_m(y)\} = \mathbb{E} \prod_{i=1}^m \exp\left\{ \frac{s}{m}(p_{\infty,\infty}^*(X_i, y) - 1) \right\}$$

$$\leq \prod_{i=1}^{m} \exp\left(\frac{s^2 C}{8m^2}\left(\frac{R}{\varepsilon}\right)^{2d_\nu}\right)$$

$$\leq \exp\left(\frac{C}{4m}\left(\frac{R}{\varepsilon}\right)^{2d_\nu}\cdot\frac{s^2}{2}\right).$$

Consequently, $U_m(y)$ is subgaussian with variance proxy $\frac{C}{4m}\left(\frac{R}{\varepsilon}\right)^{2d_\nu}$. Orlicz $\psi_2$-norm is defined for a random variable $\xi$ by

$$\|\xi\|_{\psi_2} = \inf\left\{t > 0 : \mathbb{E}e^{|\xi|^2/t^2} \leq 2\right\}.$$

Chen et al. (2023, Theorem 3.7 and Theorem 4.1) with $\varphi(x) := x^2/2$, $p = 2$ and $d(y, y') := \|U_m(y) - U_m(y')\|_{\psi_2}$ imply the following estimates:

$$\left(\mathbb{E}\sup_{y\in\operatorname{supp}(\nu)}U_m(y)^2\right)^{1/2} \lesssim \int_0^\infty \sqrt{\log\mathcal{N}(\operatorname{supp}(\nu), d, \delta)}\,d\delta + \sup_{y,y'\in\operatorname{supp}(\nu)}d(y, y'). \tag{23}$$

Let $Z_i := p_{\infty,\infty}^*(X_i, y) - p_{\infty,\infty}^*(X_i, y')$. Since each $Z_i$ is an independent, zero-mean random variable, by Vershynin (2018, Proposition 2.6.1) $d(y, y')$ is bounded above as follows:

$$d(y, y') = \left\|\mu_m\left(p_{\infty,\infty}^*(\cdot, y) - p_{\infty,\infty}^*(\cdot, y')\right)\right\|_{\psi_2}$$

$$= \frac{1}{m}\left\|\sum_{i=1}^{m}Z_i\right\|_{\psi_2}$$

$$\lesssim \frac{1}{m}\left(\sum_{i=1}^{m}\|Z_i\|_{\psi_2}^2\right)^{1/2}.$$

For each $Z_i$ we have $\|\cdot\|_{\psi_2} \lesssim \|\cdot\|_{L^\infty(\mu)}$, hence

$$\|Z_i\|_{\psi_2} = \left\|p_{\infty,\infty}^*(X_i, y) - p_{\infty,\infty}^*(X_i, y')\right\|_{\psi_2} \lesssim \left\|p_{\infty,\infty}^*(\cdot, y) - p_{\infty,\infty}^*(\cdot, y')\right\|_{L^\infty(\mu)}.$$

By $|e^a - e^b| \leq e^{a\vee b}|a - b|$ and Lemma 4, 6, for any $x$,

$$\left|p_{\infty,\infty}^*(x, y) - p_{\infty,\infty}^*(x, y')\right| \leq \left\|p_{\infty,\infty}^*\right\|_{L^\infty(\mu\otimes\nu)}\left|\log p_{\infty,\infty}^*(x, y) - \log p_{\infty,\infty}^*(x, y')\right|$$

$$\lesssim \left(\frac{R}{\varepsilon}\right)^{d_\nu}\frac{4R}{\varepsilon}\|y - y'\|_2$$

$$\lesssim \left(\frac{R}{\varepsilon}\right)^{d_\nu+1}\|y - y'\|_2.$$

Therefore,

$$d(y, y') \lesssim \frac{1}{m}\left(\sum_{i=1}^{m}\left(\frac{R}{\varepsilon}\right)^{2d_\nu+2}\|y - y'\|_2^2\right)^{1/2}$$

$$= \frac{1}{m}\left(\frac{R}{\varepsilon}\right)^{d_\nu+1}\cdot\sqrt{m}\,\|y - y'\|_2$$

$$= \frac{1}{\sqrt{m}}\left(\frac{R}{\varepsilon}\right)^{d_\nu+1}\|y - y'\|_2. \tag{24}$$

This concludes $\mathcal{N}(\operatorname{supp}(\nu), d, r) \leq \mathcal{N}\left(\operatorname{supp}(\nu), \|\cdot\|_2, \frac{r}{A}\right)$ ($A = \frac{C}{\sqrt{m}}\left(\frac{R}{\varepsilon}\right)^{d_\nu+1}$ where $C$ is a constant hidden in the above inequality).

We evaluate the first term in (23) by using Proposition 43 in Stromme (2023b) that shows there exists $0 < c_\nu \leq R$ such that, for any $0 < \delta \leq c_\nu$, the covering number satisfies $\delta^{-d_\nu} \lesssim \mathcal{N}(\operatorname{supp}(\nu), \|\cdot\|_2, \delta) \lesssim \delta^{-d_\nu}$.

$$\int_0^\infty \sqrt{\log\mathcal{N}(\operatorname{supp}(\nu), d, r)}\,dr$$

$$\lesssim \int_0^\infty \sqrt{\log \mathcal{N}\left(\mathrm{supp}(\nu), \|\cdot\|_2, \frac{r}{A}\right)}\, dr \qquad \left(A := \frac{C}{\sqrt{m}}\left(\frac{R}{\varepsilon}\right)^{d_\nu+1}\right)$$

$$= A\int_0^{2R} \sqrt{\log \mathcal{N}\left(\mathrm{supp}(\nu), \|\cdot\|_2, \delta\right)}\, d\delta \qquad \left(\delta := \frac{r}{A}\right)$$

$$= A\int_0^{c_\nu} \sqrt{\log \mathcal{N}\left(\mathrm{supp}(\nu), \|\cdot\|_2, \delta\right)}\, d\delta + A\int_{c_\nu}^{2R} \sqrt{\log \mathcal{N}\left(\mathrm{supp}(\nu), \|\cdot\|_2, \delta\right)}\, d\delta$$

$$\lesssim A\int_0^{c_\nu} \sqrt{\log \delta^{-d_\nu}}\, d\delta + A(2R - c_\nu)\sqrt{\log c_\nu^{-d_\nu}}$$

$$\lesssim A\sqrt{d_\nu}\int_0^\infty u^{1/2}e^{-u}\, du + AR\sqrt{d_\nu} \qquad \left(u := \log \frac{1}{\delta}\right)$$

$$\lesssim AR$$

$$\lesssim \frac{R}{\sqrt{m}}\left(\frac{R}{\varepsilon}\right)^{d_\nu+1}, \tag{25}$$

where $\lesssim$ also hides the intrinsic dimensionality $d_\nu$.

From (24), the second term of (23) is bounded as

$$\sup_{y,y'\in\mathrm{supp}(\nu)} d(y,y') \lesssim \sup_{y,y'\in\mathrm{supp}(\nu)} \frac{1}{\sqrt{m}}\left(\frac{R}{\varepsilon}\right)^{d_\nu+1}\|y-y'\|_2 \lesssim \frac{R}{\sqrt{m}}\left(\frac{R}{\varepsilon}\right)^{d_\nu+1}. \tag{26}$$

Combining the results of (26) and (25) completes the proof.

$$\mathbb{E}\sup_{y\in\mathrm{supp}(\nu)} U_m(y)^2 \lesssim \left(\frac{R}{\sqrt{m}}\left(\frac{R}{\varepsilon}\right)^{d_\nu+1} + \frac{R}{\sqrt{m}}\left(\frac{R}{\varepsilon}\right)^{d_\nu+1}\right)^2 \lesssim \frac{R^2}{m}\left(\frac{R}{\varepsilon}\right)^{2d_\nu+2}.$$

$$\square$$

**Lemma 9.** *Let* $V_n(x) = (\nu_n - \nu)\left(p^*_{\infty,\infty}(x,\cdot)\right)$. *Then the following holds:*

$$\mathbb{E}\sup_{x\in\mathrm{supp}(\mu)} V_n(x)^2 \lesssim \frac{dR^2}{n}\left(\frac{R}{\varepsilon}\right)^{2d_\nu+2}.$$

*Proof.* Lemma 9 can be established by a minor modification of the proof of Lemma 8. The only difference is the evaluation of the covering number: the bound over $\mathrm{supp}(\mu)$ is replaced by that over $B(0,R)$. This substitution introduces a dependence on the ambient dimension $d$. $\square$

**Lemma 10.** *It follows that*

$$\mathbb{E}\|f^*_{\infty,\infty} - f^*_{m,n}\|^2_{L^\infty(\mu)} \lesssim \left(\varepsilon^2 + \frac{dR^8}{\varepsilon^2}\right)\cdot\left(\frac{R}{\varepsilon}\right)^{7d_\nu+4}\cdot\left(\frac{1}{m}+\frac{1}{n}\right).$$

*Proof.* Let

$$\bar{f}_{\infty,n}(x) := -\varepsilon\log\left(\frac{1}{n}\sum_{j=1}^n \exp_\varepsilon(g^*_{\infty,\infty}(Y_j) - c(x,Y_j))\right).$$

Using $\bar{f}_{\infty,n}$, we evaluate $\|f^*_{\infty,\infty} - f^*_{m,n}\|^2_{L^\infty(\mu)}$ as follows.

$$\mathbb{E}\|f^*_{\infty,\infty} - f^*_{m,n}\|^2_{L^\infty(\mu)} \lesssim \mathbb{E}\|f^*_{\infty,\infty} - \bar{f}_{\infty,n}\|^2_{L^\infty(\mu)} + \mathbb{E}\|\bar{f}_{\infty,n} - f^*_{m,n}\|^2_{L^\infty(\mu)}. \tag{27}$$

For the first term, the Schrödinger system (5) implies

$$f^*_{\infty,\infty}(x) - \bar{f}_{\infty,n}(x) = \varepsilon\log\left(\frac{1}{n}\exp_\varepsilon(f^*_{\infty,\infty}(x))\sum_{j=1}^n \exp_\varepsilon(g^*_{\infty,\infty}(Y_j) - c(x,Y_j))\right)$$

$$= \varepsilon \log \left( \frac{1}{n} \sum_{j=1}^{n} p_{\infty,\infty}^*(x, Y_j) \right)$$

$$= \varepsilon \log \left( 1 + V_n(x) \right),$$

where $V_n(x) := (\nu_n - \nu)(p_{\infty,\infty}^*(x, \cdot))$ and we used $\nu(p_{\infty,\infty}^*(x, \cdot)) = 1$.

For any $\eta \in (0, 1)$, the following holds:

$$\mathbb{E}\|f_{\infty,\infty}^*(x) - \bar{f}_{\infty,n}(x)\|_{L^\infty(\mu)}^2 = \varepsilon^2 \mathbb{E} \sup_{x \in \mathrm{supp}(\mu)} |\log(1 + V_n(x))|^2$$

$$= \varepsilon^2 \mathbb{E} \left[ \mathbb{1}[\sup_{x \in \mathrm{supp}(\mu)} |V_n| < \eta] \sup_{x \in \mathrm{supp}(\mu)} |\log(1 + V_n(x))|^2 \right]$$

$$+ \varepsilon^2 \mathbb{E} \left[ \mathbb{1}[\sup_{x \in \mathrm{supp}(\mu)} |V_n| \geq \eta] \sup_{x \in \mathrm{supp}(\mu)} |\log(1 + V_n(x))|^2 \right].$$
(28)

For the first term in (28), the mean value theorem implies that for $|u| < 1$ we have $|\log(1 + u)| \leq \frac{|u|}{1-|u|}$. Hence,

$$\mathbb{E} \left[ \mathbb{1}[\sup_{x \in \mathrm{supp}(\mu)} |V_n| < \eta] \sup_{x \in \mathrm{supp}(\mu)} |\log(1 + V_n(x))|^2 \right] \leq \mathbb{E} \sup_{x \in \mathrm{supp}(\mu)} \frac{|V_n(x)|^2}{(1 - \eta)^2}.$$

For the second term in (28), since $\exp_\varepsilon(-10R^2) \leq p_{\infty,\infty}^* \leq \exp_\varepsilon(10R^2)$, an application of Markov's inequality yields

$$\mathbb{E} \left[ \mathbb{1}[\sup_{x \in \mathrm{supp}(\mu)} |V_n| \geq \eta] \sup_{x \in \mathrm{supp}(\mu)} |\log(1 + V_n(x))|^2 \right] \lesssim \frac{R^4}{\varepsilon^2} \mathbb{P}(\sup_{x \in \mathrm{supp}(\mu)} |V_n(x)| \geq \eta)$$

$$\leq \frac{R^4}{\varepsilon^2} \frac{\mathbb{E} \sup_{x \in \mathrm{supp}(\mu)} |V_n(x)|^2}{\eta^2}.$$

Setting $\eta = 1/2$ and applying Lemma 9 yields

$$\mathbb{E}\|f_{\infty,\infty}^*(x) - \bar{f}_{\infty,n}(x)\|_{L^\infty(\mu)}^2 \lesssim \left( 1 + \frac{R^4}{\varepsilon^2} \right) \mathbb{E} \sup_{x \in \mathrm{supp}(\mu)} |V_n(x)|^2$$

$$\lesssim \frac{\varepsilon^2 + dR^4}{n} \left( \frac{R}{\varepsilon} \right)^{2d_\nu + 4}.$$
(29)

For the second term in (27), Eq. (6.4) in Stromme (2023b) implies

$$|\bar{f}_{\infty,n}(x) - f_{m,n}^*(x)| \lesssim \|g_{\infty,\infty}^* - g_{m,n}^*\|_{L^2(\nu_n)} \sup_{z \in \mathrm{supp}(\nu)} \nu_n \left( B\left( z, \frac{\varepsilon}{2R} \right) \right)^{-1/2}.$$

Let $\mathcal{E}$ denote the event on which Lemma 5 holds. From Lemma 3 and $\mathbb{P}[\mathcal{E}^C] \leq \frac{1}{n}$, it follows that

$$\mathbb{E}\|f_{\infty,\infty}^* - f_{m,n}^*\|_{L^\infty(\mu)}^2 \lesssim \mathbb{E} \left[ \mathbb{1}[\mathcal{E}]\|f_{\infty,\infty}^* - f_{m,n}^*\|_{L^\infty(\mu)}^2 \right] + \frac{R^2}{n}.$$

Combining this with Lemma 7 and Lemma 5 yields

$$\mathbb{E}\|f_{\infty,\infty}^* - f_{m,n}^*\|_{L^\infty(\mu)}^2 \lesssim \mathbb{E} \left[ \mathbb{1}[\mathcal{E}]\|f_{\infty,\infty}^* - f_{m,n}^*\|_{L^\infty(\mu)}^2 \right] + \frac{R^2}{n}$$

$$\lesssim \mathbb{E} \left[ \mathbb{1}[\mathcal{E}] \left( \|f_{\infty,\infty}^* - \bar{f}_{\infty,n}\|_{L^\infty(\mu)}^2 + \|\bar{f}_{\infty,n} - f_{m,n}^*\|_{L^\infty(\mu)}^2 \right) \right] + \frac{R^2}{n}$$

$$\lesssim \frac{\varepsilon^2 + dR^4}{n} \left( \frac{R}{\varepsilon} \right)^{2d_\nu + 4} + \left( \frac{R}{\varepsilon} \right)^{d_\nu} \mathbb{E}\|g_{\infty,\infty}^* - g_{m,n}^*\|_{L^2(\nu_n)}^2 + \frac{R^2}{n}$$

$$\lesssim \left( \varepsilon^2 + \frac{dR^8}{\varepsilon^2} \right) \cdot \left( \frac{R}{\varepsilon} \right)^{7d_\nu + 4} \cdot \left( \frac{1}{m} + \frac{1}{n} \right).$$

$\square$

**Lemma 11.** *It follows that*

$$\mathbb{E}\|g^*_{\infty,\infty} - g^*_{m,n}\|^2_{L^\infty(\nu)} \lesssim \left(\varepsilon^2 + \frac{dR^8}{\varepsilon^2}\right) \cdot \left(\frac{R}{\varepsilon}\right)^{7d_\nu + 4} \cdot \left(\frac{1}{m} + \frac{1}{n}\right).$$

*Proof.* Let

$$\bar{g}_{m,\infty}(y) := -\varepsilon \log\left(\frac{1}{m} \sum_{i=1}^m \exp_\varepsilon(f^*_{\infty,\infty}(X_i) - c(X_i, y))\right).$$

Using $\bar{g}_{m,\infty}$, we evaluate $\|g^*_{\infty,\infty} - g^*_{m,n}\|^2_{L^\infty(\nu)}$ as follows.

$$\mathbb{E}\|g^*_{\infty,\infty} - g^*_{m,n}\|^2_{L^\infty(\nu)} \lesssim \mathbb{E}\|g^*_{\infty,\infty} - \bar{g}_{m,\infty}\|^2_{L^\infty(\nu)} + \mathbb{E}\|\bar{g}_{m,\infty} - g^*_{m,n}\|^2_{L^\infty(\nu)}. \tag{30}$$

For the first term, the Schrödinger system (5) implies

$$g^*_{\infty,\infty}(y) - \bar{g}_{m,\infty}(y) = \varepsilon \log\left(\frac{1}{m} \exp_\varepsilon(g^*_{\infty,\infty}(y)) \sum_{i=1}^m \exp_\varepsilon(f^*_{\infty,\infty}(X_i) - c(X_i, y))\right)$$

$$= \varepsilon \log\left(\frac{1}{m} \sum_{i=1}^m p^*_{\infty,\infty}(X_i, y)\right)$$

$$= \varepsilon \log\left(1 + U_m(y)\right),$$

where $U_m(y) := (\mu_m - \mu)(p^*_{\infty,\infty}(\cdot, y))$ and used $\mu(p^*_{\infty,\infty}(\cdot, y)) = 1$.

For any $\eta \in (0, 1)$, the following holds:

$$\mathbb{E}\|g^*_{\infty,\infty}(y) - \bar{g}_{m,\infty}(y)\|^2_{L^\infty(\nu)} = \varepsilon^2 \mathbb{E} \sup_{y \in \mathrm{supp}(\nu)} \log(1 + U_m(y))^2$$

$$= \varepsilon^2 \mathbb{E}\left[\mathbb{1}[\sup_{y \in \mathrm{supp}(\nu)} |U_m| < \eta] \sup_{y \in \mathrm{supp}(\nu)} |\log(1 + U_m(y))|^2\right]$$

$$+ \varepsilon^2 \mathbb{E}\left[\mathbb{1}[\sup_{y \in \mathrm{supp}(\nu)} |U_m| \geq \eta] \sup_{y \in \mathrm{supp}(\nu)} |\log(1 + U_m(y))|^2\right]. \tag{31}$$

For the first term in (31), the mean value theorem implies that for $|u| < 1$ we have $|\log(1 + u)| \leq \frac{|u|}{1-|u|}$. Hence,

$$\mathbb{E}\left[\mathbb{1}[\sup_{y \in \mathrm{supp}(\nu)} |U_m| < \eta] \sup_{y \in \mathrm{supp}(\nu)} |\log(1 + U_m(y))|^2\right] \leq \mathbb{E} \sup_{y \in \mathrm{supp}(\nu)} \frac{|U_m(y)|^2}{(1 - \eta)^2}$$

For the second term in (31), since $\exp_\varepsilon(-10R^2) \leq p^*_{\infty,\infty} \leq \exp_\varepsilon(10R^2)$, an application of Markov's inequality yields

$$\mathbb{E}\left[\mathbb{1}[\sup_{y \in \mathrm{supp}(\nu)} |U_m| \geq \eta] \sup_{y \in \mathrm{supp}(\nu)} |\log(1 + U_m(y))|^2\right] \lesssim \frac{R^4}{\varepsilon^2} \mathbb{P}(\sup_{y \in \mathrm{supp}(\nu)} |U_m(y)| \geq \eta)$$

$$\leq \frac{R^4}{\varepsilon^2} \frac{\mathbb{E}\sup_{y \in \mathrm{supp}(\nu)} |U_m(y)|^2}{\eta^2}.$$

Setting $\eta = 1/2$ and applying Lemma 8 yields

$$\mathbb{E}\|g^*_{\infty,\infty}(y) - \bar{g}_{m,\infty}(y)\|^2_{L^\infty(\nu)} \lesssim \left(1 + \frac{R^4}{\varepsilon^2}\right) \mathbb{E} \sup_{y \in \mathrm{supp}(\nu)} |U_m(y)|^2$$

$$\lesssim \frac{\varepsilon^2 + R^4}{m}\left(\frac{R}{\varepsilon}\right)^{2d_\nu + 4}. \tag{32}$$

For the second term of (30), since the gradient of the *log-sum-exp* function is the softmax and its $\ell_1$-norm equals 1, we obtain

$$|\bar{g}_{m,\infty}(y) - g^*_{m,n}(y)| \leq \|f^*_{\infty,\infty} - f^*_{m,n}\|_{L^\infty(\mu_m)}. \tag{33}$$

Therefore, by Lemma 10, we have

$$\mathbb{E}\|\bar{g}_{m,\infty} - g^*_{m,n}\|^2_{L^\infty(\nu)} \leq \mathbb{E}\|f^*_{\infty,\infty} - f^*_{m,n}\|^2_{L^\infty(\mu_m)}$$

$$\lesssim \left(\varepsilon^2 + \frac{dR^8}{\varepsilon^2}\right) \cdot \left(\frac{R}{\varepsilon}\right)^{7d_\nu+4} \cdot \left(\frac{1}{m} + \frac{1}{n}\right).$$

Combining the above results, we obtain

$$\mathbb{E}\|g^*_{\infty,\infty} - g^*_{m,n}\|^2_{L^\infty(\nu)} \lesssim \left(\varepsilon^2 + \frac{dR^8}{\varepsilon^2}\right) \cdot \left(\frac{R}{\varepsilon}\right)^{7d_\nu+4} \cdot \left(\frac{1}{m} + \frac{1}{n}\right). \tag{34}$$

$\square$

**Lemma 12** (Lipschitzness of softmax). *For $\lambda > 0$ and $d \geq 2$, define $\sigma_\lambda : \mathbb{R}^d \to \Delta_d$ by*

$$\big(\sigma_\lambda(x)\big)_i = \left(\frac{\exp(\lambda x_i)}{\sum_{k=1}^d \exp(\lambda x_k)}\right)_i \qquad (i = 1, \dots, d).$$

*Then for any $1 \leq p, q \leq \infty$,*

$$\|\sigma_\lambda(x) - \sigma_\lambda(y)\|_p \leq L_{p,q}\|x - y\|_q, \qquad L_{p,q} = \lambda\, 2^{-1+1/p-1/q}.$$

*Proof.* Let $s = \sigma_\lambda(z)$. The Jacobian of $\sigma_\lambda$ at $z$ is

$$J(z) = \nabla \sigma_\lambda(z) = \lambda\big(\mathrm{diag}(s) - ss^\top\big). \tag{35}$$

By the fundamental theorem of calculus along the segment $t \mapsto y + t(x - y)$,

$$\sigma_\lambda(x) - \sigma_\lambda(y) = \int_0^1 J(y + t(x - y))\,(x - y)\,\mathrm{d}t,$$

hence, for any $1 \leq p, q \leq \infty$,

$$\|\sigma_\lambda(x) - \sigma_\lambda(y)\|_p \leq \left(\sup_z \|J(z)\|_{q \to p}\right)\|x - y\|_q. \tag{36}$$

It suffices to bound $\sup_z \|J(z)\|_{q \to p}$.

**Step 1: Decomposition.** With the standard basis $\{e_i\}_{i=1}^d$,

$$\mathrm{diag}(s) - ss^\top = \sum_{i<j} s_i s_j\,(e_i - e_j)(e_i - e_j)^\top. \tag{37}$$

Thus for any $u \in \mathbb{R}^d$,

$$J(z)u = \lambda \sum_{i<j} s_i s_j\,(u_i - u_j)(e_i - e_j),$$

and by the triangle inequality and $\|e_i - e_j\|_p = 2^{1/p}$ (with the convention $2^{1/\infty} = 1$),

$$\|J(z)u\|_p \leq \lambda\, 2^{1/p} \sum_{i<j} s_i s_j\,|u_i - u_j|. \tag{38}$$

**Step 2: Maximization over $s \in \Delta_d$.** Let $i_{\min} = \arg\min_i u_i$, $i_{\max} = \arg\max_i u_i$. Since $|u_i - u_j| \leq |u_{i_{\max}} - u_{i_{\min}}|$ and $\sum_{i<j} s_i s_j = \frac{1}{2}\big(1 - \sum_{i=1}^d s_i^2\big)$,

$$\sum_{i<j} s_i s_j\,|u_i - u_j| \leq \frac{|u_{i_{\max}} - u_{i_{\min}}|}{2}\Big(1 - \sum_{i=1}^d s_i^2\Big) \leq \frac{|u_{i_{\max}} - u_{i_{\min}}|}{4}.$$

The last inequality is tight when $s_{i_{\max}} = s_{i_{\min}} = \frac{1}{2}$ and all other $s_i = 0$. Therefore,

$$\sup_{s \in \Delta_d} \sum_{i<j} s_i s_j\,|u_i - u_j| = \frac{|u_{i_{\max}} - u_{i_{\min}}|}{4}. \tag{39}$$

Because $\sigma_\lambda(\mathbb{R}^d)$ is the interior of $\Delta_d$ and the right-hand side of (38) is continuous in $s$, the supremum over $z$ equals the supremum over $s \in \Delta_d$.

**Step 3: Maximization over $\|u\|_q = 1$.** For any $a, b \in \mathbb{R}$ and $q \geq 1$, $|a - b|^q \leq 2^{q-1}(|a|^q + |b|^q)$. With $a = u_{i_{\max}}$, $b = u_{i_{\min}}$ and $\|u\|_q = 1$,

$$|u_{i_{\max}} - u_{i_{\min}}| \ \leq \ 2^{1-1/q}\Big(|u_{i_{\max}}|^q + |u_{i_{\min}}|^q\Big)^{1/q} \ \leq \ 2^{1-1/q}, \tag{40}$$

with equality when $u$ has exactly two nonzero entries $u_{i_{\max}} = 2^{-1/q}$ and $u_{i_{\min}} = -2^{-1/q}$.

**Step 4: Combine.** From (38)–(40),

$$\sup_{z} \sup_{\|u\|_q=1} \|J(z)u\|_p \ \leq \ \lambda\, 2^{1/p} \cdot \frac{1}{4} \cdot 2^{1-1/q} = \lambda\, 2^{-1+1/p-1/q}.$$

Plug this into (36) to obtain the claim. The bound is tight by the equality cases noted in (39) and (40). $\qquad\square$

**Lemma 13.** *For any probability vectors $u, v \in \Delta_d$,*

$$\|u - v\|_1 \leq \frac{1}{2}\, d_{\mathrm{Hilb}}(u, v). \tag{41}$$

*Proof.* By Cohen and Fausti (2024)[Theorem 5.1],

$$\|u - v\|_1 \leq 2 \tanh\!\left(\frac{d_{\mathrm{Hilb}}(u, v)}{4}\right). \tag{42}$$

Since $\tanh(x) \leq x$ for $x \geq 0$ and $d_{\mathrm{Hilb}}(u, v) \geq 0$, letting $x = \frac{d_{\mathrm{Hilb}}(u,v)}{4}$ gives

$$\|u - v\|_1 \leq 2 \tanh\!\left(\frac{d_{\mathrm{Hilb}}(u, v)}{4}\right) \tag{42}$$

$$\leq 2\left(\frac{d_{\mathrm{hilb}}(u, v)}{4}\right) = \frac{1}{2}\, d_{\mathrm{hilb}}(u, v). \tag{43}$$

This proves (41). $\qquad\square$

**Lemma 14.** *For any $\mathbf{u}, \mathbf{v} \in \mathbb{R}_+^d$, the following holds:*

$$d_{\mathrm{Hilb}}(\mathbf{u}, \mathbf{v}) = d_{\mathrm{Hilb}}(\mathbf{v}, \mathbf{u}),$$
$$d_{\mathrm{Hilb}}(\mathbf{u}, c\mathbf{v}) = d_{\mathrm{Hilb}}(\mathbf{u}, \mathbf{v}) \quad \forall c > 0,$$
$$d_{\mathrm{Hilb}}(\mathbf{c} \odot \mathbf{u}, \mathbf{c} \odot \mathbf{v}) = d_{\mathrm{Hilb}}(\mathbf{u}, \mathbf{v}) \quad \forall \mathbf{c} \in \mathbb{R}_+^d.$$

*Here, $\odot$ denotes the element-wise product.*

*Proof.* The first two properties are established in (Lemmens and Nussbaum, 2013, Lemma 2.1). For the third property, we have

$$d_{\mathrm{Hilb}}(\mathbf{c} \odot \mathbf{u}, \mathbf{c} \odot \mathbf{v}) = \log\left(\max_i \frac{c_i u_i}{c_i v_i} \cdot \max_i \frac{c_i v_i}{c_i u_i}\right)$$
$$= \log\left(\max_i \frac{u_i}{v_i} \cdot \max_i \frac{v_i}{u_i}\right)$$
$$= d_{\mathrm{Hilb}}(\mathbf{u}, \mathbf{v}).$$

$\qquad\square$

## B.2 Lemma used for the drift estimation using neural networks

We provide an alternative expression of the Sinkhorn bridge drift estimator as a Markovian projection. Recall that $W^\varepsilon_{|x_0,x_1}$ is the Brownian bridge connecting $x_0$ and $x_1$, so its marginal distribution at time $t \in [0, 1]$ is a Gaussian distribution $W^\varepsilon_{t|x_0,x_1} = \mathcal{N}\big((1 - t)x_0 + tx_1, \varepsilon t(1 - t)I_d\big)$. For any distributions $\mu', \nu' \in \mathcal{P}(\mathbb{R}^d)$, and dual potentials $f \in L^1(\mu')$, $g \in L^1(\nu')$, we define

$$d\pi(x_0, x_1) = \exp\left(\frac{f(x_0) + g(x_1) - \frac{1}{2}\|x_0 - x_1\|_2^2}{\varepsilon}\right) d(\mu' \otimes \nu')(x_0, x_1).$$

We here only consider the case where $\pi$ becomes a joint distribution $\pi \in \mathcal{P}(\mathbb{R}^d \times \mathbb{R}^d)$, that is, $\int d\pi(x_0, x_1) = 1$. Using $\pi_{m,n}^{(k)}$, we define the mixture of Brownian bridge (reciprocal process): $\Pi = \int W_{|x_0,x_1}^{\varepsilon} d\pi(x_0, x_1) \in \mathcal{P}(\Omega)$.

**Lemma 15.** *Under the above setting, it follows that*

$$b(x_t, t) := \mathbb{E}_{X \sim \Pi} \left[ \frac{X_1 - x_t}{1 - t} \,\middle|\, X_t = x_t \right]$$

$$= \frac{1}{1 - t} \left( -x_t + \frac{\mathbb{E}_{X_1 \sim \nu'}[\gamma(X_1, x_t) X_1]}{\mathbb{E}_{X_1 \sim \nu'}[\gamma(X_1, x_t)]} \right),$$

*where $\gamma(x_1, x_t) = \exp\left( \left( g(x_1) - \frac{1}{2(1-t)} \|x_1 - x_t\|^2 \right) / \varepsilon \right)$.*

*Proof.* Given $x_t$ ($t < 1$), we denote by $\Pi(x_1 | X_t = x_t)$ the conditional density of $\Pi$ with respect to Lebesgue measure $dx_1$. Note that the marginal density $\Pi_{t1}$ of $\Pi$ on time points $t$ and $1$ with respect to $dx_t dx_1$ satisfies that for a measurable $A \subset \mathbb{R}^d$

$$\int_A \Pi_{t1}(x_t, x_1)\, dx_1 = \int W_{t|x_0,x_1}^{\varepsilon}(x_t) \pi(dx_0, A).$$

Therefore, we get

$$\Pi(x_1 | X_t = x_t)\, dx_1 \propto \left\{ \int \exp\left( -\frac{1}{2\varepsilon t(1-t)} \|x_t - (1-t)x_0 - tx_1\|^2 \right) \right.$$

$$\left. \cdot \exp\left( \frac{g(x_1) - \frac{1}{2}\|x_0 - x_1\|_2^2}{\varepsilon} \right) d\mu'(x_0) \right\} \cdot d\nu'(x_1)$$

$$\propto \left\{ \int \exp\left( -\frac{1}{2\varepsilon t(1-t)} (t^2 \|x_1\|^2 - 2t x_t^\top x_1 + 2t(1-t)x_0^\top x_1) \right) \right.$$

$$\left. \cdot \exp\left( \frac{g(x_1)}{\varepsilon} - \frac{1}{2\varepsilon}(\|x_1\|^2 - 2x_0^\top x_1) \right) d\mu'(x_0) \right\} \cdot d\nu'(x_1)$$

$$= \exp\left( \frac{g(x_1)}{\varepsilon} - \frac{1}{2\varepsilon(1-t)}(\|x_1\|^2 - 2x_t^\top x_1) \right) d\nu'(x_1)$$

$$\propto \exp\left( \frac{1}{\varepsilon} \left( g(x_1) - \frac{1}{2(1-t)} \|x_1 - x_t\|^2 \right) \right) d\nu'(x_1).$$

This means

$$\Pi(x_1 | X_t = x_t)\, dx_1 = \frac{\gamma(x_1, x_t)\, d\nu'(x_1)}{\mathbb{E}_{X_1 \sim \nu'}[\gamma(X_1, x_t)]}.$$

Therefore,

$$b(x_t, t) = \frac{1}{1-t} \left( -x_t + \mathbb{E}_{X \sim \Pi}[X_1 \mid X_t = x_t] \right)$$

$$= \frac{1}{1-t} \left( -x_t + \int x_1 \frac{\gamma(x_1, x_t)}{\mathbb{E}_{X_1 \sim \nu'}[\gamma(X_1, x_t)]} d\nu'(x_1) \right)$$

$$= \frac{1}{1-t} \left( -x_t + \frac{\mathbb{E}_{X_1 \sim \nu'}[\gamma(X_1, x_t) X_1]}{\mathbb{E}_{X_1 \sim \nu'}[\gamma(X_1, x_t)]} \right).$$

$\square$

## C  Omitted Experiments

### C.1  Stopping and guidance

Theorems 1 and 2 suggest that beyond a certain point additional Sinkhorn iterations are not beneficial. The total error decomposes into a "sampling error" and an "optimization error". The latter decreases

Table 1: Comparison of $\int_0^{0.9} \mathrm{MSE}_{\mathrm{sample}}(m, n, t)dt$ across intrinsic dimensions $d_\nu$.

| $d_\nu$ | $\int_0^{0.9} \mathrm{MSE}_{\mathrm{sample}}(m, n, t)dt$ |
|---|---|
| 5 | 0.031 |
| 10 | 0.138 |
| 20 | 0.562 |
| 30 | 1.178 |

only through iterations, whereas the former does not. By the triangle inequality for the total variation distance,

$$\mathbb{E}\left[\mathrm{TV}^2\left(P_{\infty,\infty}^*, P_{m,n}^k\right)\right] \lesssim \mathbb{E}\left[\mathrm{TV}^2\left(P_{\infty,\infty}^*, P_{m,n}^*\right)\right] + \mathbb{E}\left[\mathrm{TV}^2\left(P_{m,n}^*, P_{m,n}^k\right)\right].$$

Once the second term on the right-hand side (optimization error) becomes no larger than the first term (estimation error), further iterations no longer yield a meaningful reduction of the total error. Hence it is reasonable to choose the stopping point as the smallest $k$ such that

$$\mathbb{E}\left[\mathrm{TV}^2\left(P_{\infty,\infty}^*, P_{m,n}^*\right)\right] \geq \mathbb{E}\left[\mathrm{TV}^2\left(P_{m,n}^*, P_{m,n}^k\right)\right].$$

From Theorems 1 and 2, a sufficient condition for this is

$$k \gtrsim \frac{\log\left(\frac{B}{m} + \frac{A+B}{n}\right)}{\log\left(\tanh\left(\frac{R^2}{\varepsilon}\right)\right)}, \quad A = \frac{1}{\tau(1-\tau)^{d_\nu+1}\varepsilon^{d_\nu-2}R^4}, \quad B = \left(\frac{\varepsilon^4}{R^6} + dR^2\right)\left(\frac{R}{\varepsilon}\right)^{9d_\nu+4}.$$

Since $\log\left(\tanh\left(\frac{R^2}{\varepsilon}\right)\right) < 0$, iterations are beneficial only when $\frac{B}{m} + \frac{A+B}{n} < 1$. Equivalently, unless the sample size is large enough to make the numerator negative—roughly $(m \wedge n) \approx (A \vee B)$—the reduction achievable by iterations is smaller than the estimation error, leading to wasted computational resources.

In Figure 3, the fact that $b_{m,n}^{(1)}$ already captures the shape of $\nu_n$ to a certain extent is consistent with this reasoning. Increasing $k$ becomes meaningful only when $\mu_m$ and $\nu_n$ are sufficiently close to the true distributions $\mu$ and $\nu$, namely when $m$ and $n$ are sufficiently large.

## C.2 Intrinsic dimension

Theorem 1 shows that the orders of the orders of $R$ and $\varepsilon$ depend on the intrinsic dimension $d_\nu$ rather than the ambient dimension $d$. In this section, we verify that the error varies with $d_\nu$ under the following setup.

We fix the ambient dimension at $d = 50$, the regularization strength at $\varepsilon = 0.5$, the evaluation-interval endpoint at $\tau = 0.9$, and the sample sizes at $m = n = 10000$. The distribution $\mu$ is sampled uniformly from the unit hypercube $[0, 1]^d$, while $\nu$ is sampled uniformly from the $d_\nu$-dimensional manifold embedded in the unit sphere of radius 1,

$$\left\{ x \in \mathbb{R}^d \mid \|x\|_2 = 1, \ \forall i > d_\nu : \ x_i = 0 \right\}.$$

By varying $d_\nu$, we examine whether the convergence behavior of the estimation error with respect to the sample size depends on the manifold dimension.

Specifically, for each $d_\nu \in \{5, 10, 20, 30\}$ we compare

$$\int_0^{0.9} \mathrm{MSE}_{\mathrm{sample}}(m, n, t)\, \mathrm{d}t.$$

The results are reported in Table 1. An increasing trend of the error with larger manifold dimension is observed.

## C.3 The role of epsilon

We examine the numerical stability of the regularization parameter $\varepsilon$ in the Sinkhorn algorithm and provide selection guidelines in relation to the sample sizes $(m, n)$. Using the same setup as

Table 2: Representative $\varepsilon$ achieving the target error $\delta = 1.0$ for varying $m \wedge n$ (same setup as Experiments 4.1).

| $m \wedge n$ | $\varepsilon$ |
|---|---|
| 10 | $2.684 \times 10^8$ |
| 20 | $7.979 \times 10^6$ |
| 30 | $3.200 \times 10^1$ |
| 40 | $6.797 \times 10^{-1}$ |
| 50 | $3.291 \times 10^{-1}$ |
| 60 | $1.299 \times 10^{-1}$ |
| 70 | $2.010 \times 10^{-1}$ |
| 80 | $9.766 \times 10^{-4}$ |

Experiments 4.1, for each given $m, n$ we search for the smallest $\varepsilon$ that achieves the target error $\delta = 1.0$.

As $m \wedge n$ increases, the $\varepsilon$ required to meet the target error exhibits a two-phase behavior: a steep initial decrease followed by an asymptotically mild decay. Representative values are reported in Table 2.

By Theorem 1, the $\varepsilon$ required to satisfy the target error $\delta$ obeys

$$\varepsilon \gtrsim \max \left\{ \left( \frac{R^2}{\delta(m \wedge n)} \right)^{1/d_\nu}, R^{\frac{3d_\nu+4}{3d_\nu+2}} \left( \frac{R^2+1}{\delta(m \wedge n)} \right)^{1/(9d_\nu+6)} \right\}.$$

This indicates a steep improvement in the small-$m \wedge n$ regime followed by logarithmically slow decay, which aligns with the empirical results. Since excessively small $\varepsilon$ can cause numerical instability, using the above expression as a lower-bound guideline for selecting $\varepsilon$ is effective.

