# OpenReview forum: "Statistical Analysis of the Sinkhorn Iterations for Two-Sample Schr\"{o}dinger Bridge Estimation"
_NeurIPS.cc/2025/Conference — NeurIPS 2025 poster_

### Official Review · Reviewer_AupN · 2025-06-03

**Clarity:** 3
**Significance:** 3
**Originality:** 3
**Rating:** 4
**Confidence:** 2

**Summary:**

This paper analyzes the statistical and algorithmic behavior of Sinkhorn-based Schrödinger bridge estimators in the two-sample setting,
where both source and target distributions are accessible only via finite samples. The authors establish a total variation error bound,
quantifying how the estimator improves with more samples and Sinkhorn iterations. They further show that several popular Schrödinger
bridge solvers (e.g., [SF]²M, DSBM-IMF, BM2, lightSB) are equivalent to the Sinkhorn bridge and thus inherit the same guarantees. Finally,
they propose a neural network approximation for the drift to reduce simulation cost. Experiments validate both theoretical rates and the
efficiency of the learned drift.

**Questions:**

1. How does the proposed Sinkhorn bridge estimator perform compared to standard reverse diffusion interpolations or Schrödinger bridge
solvers that use learned forward-backward dynamics, both in accuracy and computational cost?
2. The neural network drift approximation reduces per-step cost from $\mathcal{O}(n)$ to $\mathcal{O}(1)$, but does this come at the cost of
accuracy?

**Ethical Concerns:**

["NO or VERY MINOR ethics concerns only"]

**Final Justification:**

The authors have addressed my concerns thoroughly, and I have no further issues. This paper presents strong theoretical results and has been well-received by all reviewers. However, due to my unfamiliarity with the relevant theory, I will keep my score and confidence level unchanged. I appreciate the efforts of both the authors and the AC—apologies for not being able to fully engage in the rebuttal and adjust my rating accordingly.

**Limitations:**

Yes.

**Paper Formatting Concerns:**

I do not find any.

**Quality:**

3

**Strengths And Weaknesses:**

**Strengths:**
1. Provides a sharp finite-sample error bound $\mathcal{O}(1/m + 1/n + r^{2k})$ for Sinkhorn-based Schrödinger bridge estimation,
covering both sample and iteration complexity.
2. Extends prior work from the one-sample setting to the more general and practical two-sample setting.
3. Unifies several existing SB methods (\[SF]²M, DSBM-IMF, BM2, lightSB) under the Sinkhorn bridge framework, giving them new
theoretical guarantees.
4. Introduces a neural network–based drift approximation that significantly reduces simulation cost from $\mathcal{O}(n)$ to $\mathcal{O}
(1)$ per step.
5. Empirical results validate both the statistical convergence and the efficiency of the neural approximation.

**Weaknesses:**
1. Experimental results are primarily demonstrated in low-to-moderate dimensions; scalability to high-dimensional applications remains to be tested.

---

> ### Author Rebuttal · Authors · 2025-07-31
>
> We thank the reviewer for reading our paper.
>
> **Accuracy and computational cost of Sinkhorn bridge**
>
> - Accuracy: The Sinkhorn bridge estimator performs well in terms of accuracy. In fact, [1] observed that it outperforms Schrödinger bridge solvers based on learned forward–backward dynamics under the experimental setting of [2]. For details, see Table 1 in [1].
>
>
> - Computational cost: The Sinkhorn Bridge requires the coupling $\pi_{m,n}$ which can be obtained using Sinkhorn iterations. These iterations converge exponentially fast and have a per-iteration cost of $mn$. In the sampling phase, the Sinkhorn bridge applies Euler–Maruyama discretization, which incurs a cost of $O(n)$ per discretization step.
> Therefore, generating a new sample from the estimated target distribution costs $O(nK)$, where $K$ is the number of discretization steps. To reduce the inference cost, we also proposed a neural network-based approximation of the drift.
>
>
> [1] Pooladian, A.-A. and Niles-Weed, J. "Plug-in estimation of Schrödinger bridges." arXiv:2408.11686, 2024.
>
> [2] Gushchin, N., et al. Building the bridge of Schr ̈odinger: A continuous entropic optimal transport benchmark. NeurIPS, 2023.
>
> **Neural network based drift approximation**
>
> We note that Algorithm 1 performs online SGD to minimize $\mathcal{L}(b)$ (line 210), and this is feasible since $\mathcal{L}(b)$ is tractable. Specifically, the expectation in $\mathcal{L}(b)$ is taken with respect to a mixture of Brownian bridges defined by the transportation plan $\pi$, from which we can sample to perform SGD. As a result, there are approximation errors (due to model expressivity) and optimization errors (due to finite training iterations), but with sufficient network capacity and training time, the NN-based estimator will converge to the Sinkhorn bridge drift, which is the minimizer of $\mathcal{L}(b)$.

---

### Official Review · Reviewer_Lkjn · 2025-06-25

**Clarity:** 3
**Significance:** 2
**Originality:** 3
**Rating:** 5
**Confidence:** 3

**Summary:**

The paper extends the analysis of the plug-in Schrödinger bridge estimator proposed in [1] in several ways:
* The analysis is generalized to the two-sample setting.
* The convergence rate with respect to the sample size $n$ is improved compared to [1] (Theorem 1).
* The analysis incorporates the number of Sinkhorn iterations as a parameter, which was absent in [1] (Theorem 2).

In addition, the paper provides an alternative computational implementation using neural networks, which avoids the computation at every discretization step (Algorithm 1).


[1] Aram-Alexandre Pooladian, Jonathan Niles-Weed. Plug-in estimation of Schrödinger bridges. Arxiv preprint https://arxiv.org/abs/2408.11686, 2024.

**Questions:**

* Minor typo: Line 196 contains a typo. It probably should be $\mathbb{E}\left[TV^2(P_{m,n}^{k,[0,\tau]}, P_{\infty,\infty}^{*,[0,\tau]})\right]$?

* As mentioned in the weaknesses section, the result from this paper seems to indicate a trade-off between $n$ and $R$. Do you think this trade-off is unavoidable? It seems that the cost is quite large.

* In Remark 1, the authors mention that the dimension $d$ can be replaced by the intrinsic dimension $d_{\nu}$, but Line 308 states that the theorem relies on the ambient dimension rather than the intrinsic dimension. Could you elaborate on this?

**Ethical Concerns:**

["NO or VERY MINOR ethics concerns only"]

**Final Justification:**

As elaborated, I found the Section 3.2 as an incremental improvement compared to [1]; while there is an improvement in the rate, I find the improvement relatively restrictive. On the other hand, I find Section 3.3 favorable, as it offers the computational advantage which [1] lacked. My initial concern on Section 3.3 was its algorithmic convergence, and summing up these points I rated the borderline accept at the first review. However, I found out that I had some misunderstanding regarding its algorithmic convergence, and as a result my concern was resolved during the discussion period.

In conclusion, I would like to rate accept to this paper.

**Limitations:**

The authors expressed limitations of the work. I have stated additional limitations that I found in the weaknesses section.

**Paper Formatting Concerns:**

I did not find any issues with the paper formatting.

**Quality:**

4

**Strengths And Weaknesses:**

### Strengths
* As mentioned in the summary, the paper extends the plug-in Schrödinger bridge estimator proposed in [1] in several ways. In particular, improving the convergence rate in $n$ and including the Sinkhorn iteration parameter in the bound are new contributions that were absent in the previous literature.

* The paper shows that the estimator proposed in [1] can be interpreted as the minimizer of a certain loss functional. Consequently, the authors propose an alternative implementation of the estimator using a neural network. This implementation is favorable, as one can use a single neural network that takes the time-step as input, rather than computing the drift term at each time-step.

### Weaknesses
* It seems that in order to improve the rate from $O(n^{-1/2})$ to $O(n^{-1})$ (assuming $m \to \infty$), one obtains a substantially worse rate with respect to $R$. This restricts the implications of the analysis, as the bound becomes significantly worse unless $R$ is very small.

* While I find the estimator proposed in Algorithm 1 interesting, I suspect that the estimator obtained from this algorithm may have different statistical guarantees compared to the plug-in estimator analyzed in the main body. The statistical error between the population loss minimizer (Line 210) and the empirical loss minimizer (Algorithm 1) should involve some dependence on $N$. This statistical error should be carefully analyzed, as the loss is non-convex and thus convergence is not straightforward.

---

> ### Author Rebuttal · Authors · 2025-07-31
>
> We thank the reviewer for carefully reading our paper.
>
> **Tradeoff between $n$ and $R$**
>
> Several recent works that established the faster convergence rate of $O(1/n)$ for EOT assumed either the boundedness of $R$ or Lipschitz continuous bounded cost (i.e, the bounded $R$ with Euclidean cost.) These results suggest a trade-off between $n$ and $R$. However, the optimal dependence on these constants remains unclear.
> Our bound exhibits exponential dependence on $R$, but we  note that such dependence is common in the literature of the entropic transport [1,2]. In fact, the term $\exp(CR^2)$ in our bound is inherited from the existing bound on the Schrödinger potential functions [2]. More recently, [3] established the polynomial bound of the form $R^d$ for $L^2$-distance between potential functions. However, translating their result for EOT into statistical bounds for the Schrödinger bridge is nontrivial and is a future research direction.
>
>
> [1] Rigollet, Philippe, and Austin J. Stromme. "On the sample complexity of entropic optimal transport." The Annals of Statistics, 2025.
>
> [2] del Barrio, Eustasio, et al. "An improved central limit theorem and fast convergence rates for entropic transportation costs." SIAM Journal on Mathematics of Data Science, 2023.
>
> [3] Stromme, A. J. "Minimum intrinsic dimension scaling for entropic optimal transport." In International Conference on Soft Methods in Probability and Statistics, 2024.
>
> **Dependence on (intrinsic) dimension**
>
> Our bound relies on the following decomposition:
> $\mathbb{E}[TV^2(P\_{\infty,\infty}^{\*,[0,\tau]},P\_{m,n}^{\*,[0,\tau]} )] \lesssim \mathbb{E}[KL(P\_{\infty,\infty}^{\*,[0,\tau]} | \bar{P}\_{\infty,n}^{[0,\tau]})] + \mathbb{E}[KL(\bar{P}\_{\infty,n}^{[0,\tau]} | P\_{m,n}^{\*, [0,\tau]}) ]$.
>
> The first term admits an upper bound of the form: $\frac{R^2}{n (1-\tau)^{d_{\nu}+2} \varepsilon^{d_{\nu}}}$ (Proposition 4.3 in [4]) when the support of $\nu$ is a $d_{\nu}$-dimensional submanifold. However, for the second term, we have not obtained a bound that depends on the intrinsic dimension $d_{\nu}$. As a result, the overall bound depends on the ambient dimension $d$ rather than $d_\nu$.
>
> [4] Pooladian, A.-A. and Niles-Weed, J. Plug-in estimation of Schrödinger bridges. arXiv:2408.11686, 2024.
>
> **Weakness 2: relationship between Algorithm 1 and plug-in estimator**
>
> We note that Algorithm 1 performs online SGD to minimize $\mathcal{L}(b)$ (line 210), and this is feasible since $\mathcal{L}(b)$ is tractable. Specifically, the expectation in $\mathcal{L}(b)$ taken with respect to a mixture of Brownian bridges defined by the transportation plan $\pi$, from which we can sample to perform SGD. As a result, Algorithm 1 converges to the plug-in estimator, and no additional statistical error is incurred if the neural network is sufficiently large. It is also important to note that $N$ denotes the mini-batch size of SGD and is unrelated to the definition of the objective $\mathcal{L}(b)$.
>
> **Minor typo**
>
> Thank you for noticing the typo. It has been corrected accordingly.

---

> > ### Comment · Reviewer_Lkjn · 2025-08-02
> >
> > I appreciate the authors' detailed response.
> >
> > 1. Based on the authors' answer, I agree improving the trade-off between $R$ and $n$ is a difficult task. I think this does not harm the novelty of the paper, while I would like to make one comment why I thought this trade-off limits the implication. In many cases, for general data, what one typically do is use the concentration inequality to bound $\max ||x_i|| < R$ with high probability, and apply the bounded case analyses. Lastly, one plugs-in the $R$ derived from the concentration inequality. However, if $R^2$ has the exponential rate, such strategy breaks in many cases, as for typical sub-Gaussian data, $\exp(R^2)$ is of the form $n/\delta$, and leads to $O(1)$ bound. In this regard, I personally find the bound of [1] more favorable in general (note they have the polynomial rate of $R$). However, it is true for the constant $R$ the proposed bound is more favorable. One suggestion is to state that the proposed bound is more favorable than [1] if the data is known to be bounded.
> >
> > 2. For Algorithm 1, even if $\mathcal{L}$ is tractable, this problem has two separate errors occurring. First, since the authors are using SGD with Monte Carlo estimated loss, there is an error between the population minimizer $b$ and obtained $b_N$ from SGD updates, w.r.t. $N$. This is what I meant by statistical error. Second, there is a neural network approximation error, which also makes the problem non-convex (this is very hard problem though). What I thought when writing the review was that at least first error can be controlled: the variance of the stochastic gradient in this problem seems tractable, so one might be able to conduct the variance control analysis as in usual SGD analysis and get the bound with respect to $N$ as well. But after finishing the review, I found out that this might be harder than I expected.

---

> ### Author Response · Authors · 2025-08-02
>
> We thank the reviewer for engaging with us in the discussion.
>
> **2: Statistical error in Algorithm 1**
>
> We would first like to clarify an important point that may have been misunderstood in the review. It seems the reviewer may have interpreted Algorithm 1 as optimizing an empirical approximation to $\mathcal{L}(b_\theta)$, possibly because we introduce the empirical approximation $L(\theta)$ in Algorithm 1. Also we stated "_SGD can be efficiently applied to minimize_ $L(\theta)=\mathcal{L}(b_\theta)$" in the submission. This is actually our typo, which is correctly "_SGD can be efficiently applied to minimize_ $\mathcal{L}(b_\theta)$"
>
> We emphasize that Algorithm 1 does not optimize empirical loss $L(\theta)$. Instead, $L(\theta)$ is used solely to define a stochastic gradient at each iteration, and the mini-batch that defines $L(\theta)$ is resampled every time. Therefore, Algorithm 1 performs SGD directly on the true objective $\mathcal{L}(b_\theta)$, and convergence to a stationary point is guaranteed even when using $N=1$, where $N$ is simply the mini-batch size in SGD.
>
> To make this clearer, let us abstract the setting using a general form: suppose the objective function is $F(\theta) = E_Z[ l(\theta, Z)]$, where $Z$ follows a continuous distribution. Then, Algorithm 1 with $N=1$ reduces to the standard online SGD update $\theta_{k+1} = \theta_k - \eta \nabla_\theta l(\theta_k, Z_k)$ where $\\{Z_k\\}_k$ are i.i.d. samples. The convergence of such SGD with small or diminishing step-size to a stationary point of $F(\theta)$ is well-established in the literature. For example, see Theorems 4.8–4.10 and related corollaries in the following reference. (Actually, the objective $F$ in these results covers the expected loss. See Eq. (4.1) in [5].)
>
> [5] L Bottou, F.E. Curtis, and J. Nocedal. Optimization Methods for Large-Scale Machine Learning
> https://arxiv.org/pdf/1606.04838
>
> As for concerns regarding the approximation error due to neural networks and global convergence, we believe these are minor. Most neural network-based diffusion models and Schrödinger bridge methods face similar challenges. That said, we note that the objective $\mathcal{L}(b)$ is (linearly) convex in $b$, so global convergence can be ensured using recent results from mean-field theory [6, 7, 8].
>
> We would be happy to address any further concerns during the discussion period.
>
> [6] L. Chizat and F. Bach. On the global convergence of gradient descent for over-parameterized models using optimal transport. NeurIPS, 2018
>
> [7] A. Nitanda, D. Wu, and T. Suzuki. Convex Analysis of the Mean Field Langevin Dynamics. AISTATS, 2022.
>
> [8] L. Chizat. Mean-field Langevin dynamics: exponential convergence and annealing. TMLR, 2022.
>
> **1: Tradeoff between $R$ and $n$**
>
> Thank you for the suggestion. Following your advice, we will include a comment to address this point in the final version.

---

> > ### Comment · Reviewer_Lkjn · 2025-08-03
> >
> > I truly appreciate the authors for their detailed feedback.
> >
> > I apologize for the confusion I made. Now I understand the authors claim, and I agree with the authors that SGD guarantees the convergence. It seems like one can proceed the analysis via standard SGD arguments for this part. And I agree that the error from neural network approximation is beyond the scope of this work.
> >
> > Given the fact that my concern on Section 3.3 was from my misunderstanding, I would like to change my overview of the work as follows:
> >
> > 1. As mentioned in my initial review, section 3.2 extends [1] in several ways: The extension to two sample testing and the improvement of the rate w.r.t. $n$. While there is a trade-off of the rate between $n$ and $R$ which limits the implication of the improvement on $n$ for general cases, there is some situations where such improvement is favorable.
> >
> > 2. Section 3.3 provides a variational formulation of the drift term and suggests the neural network implementation. I find the variational formulation strong, as in my opinion this improves the computational challenges [1] had; instead of computing the drift term at each time-step, one can use a single neural network that takes the time-step as input. In fact, it does not have to be a neural network, and instead one can take any approximation methods (e.g., dictionary learning) which are capable of approximating the drift term (I think this is also one point the authors can emphasize in my personal opinion).
> >
> > As a consequence, I decided to raise my score.

---

> > > ### Author Response · Authors · 2025-08-04
> > >
> > > We appreciate your sincere engagement in the discussion. Following your suggestions, we will emphasize more on (1) the conditions under which our two-sample statistical error analysis is favorable (i.e., when $R$ is bounded), and (2) the flexibility of our framework (Algorithm 1), which allows for arbitrary approximation methods to estimate the plug-in estimator. We believe these clarifications will highlight the benefits of our work.

---

### Official Review · Reviewer_EHkZ · 2025-07-01

**Clarity:** 4
**Significance:** 3
**Originality:** 3
**Rating:** 4
**Confidence:** 3

**Summary:**

This works investigates the statistical guarantees of the Sinkhorn bridge algorithm in the two-sample estimation setting.

**Questions:**

See above.

**Ethical Concerns:**

["NO or VERY MINOR ethics concerns only"]

**Final Justification:**

The authors clarified the relevance of the paper. I'd like to maintain my positive assessment.

**Limitations:**

See above.
I think some clarifications on the relevance to the greater AI community would be nice for general audience to appreciate the main contributions. But I should say I'm not quite familiar with the literature.

**Quality:**

4

**Strengths And Weaknesses:**

I enjoyed reading through the results. The results extend the results of Pooladian and Niles-Weed (2024) to the two-sample estimation setting. The theoretical analysis is built on the techniques of Pooladian and Niles-Weed (2024) and seems sound.

The empirical validation is also clearly presented for synthetic Gaussian settings.

Overall, I think this paper is a solid theoretical contribution to the community. I do have to say that I don't work in this field, so my overall evaluation of this paper is limited to the presentation of this paper. I do find the problem and the results great.

That said, I'm not very sure about the relevance of this paper to the NeurIPS community. I'd appreciate if the authors could provide the relevance of this result for more general audience and in their final version.

---

> ### Author Rebuttal · Authors · 2025-07-30
>
> We thank the reviewer for reading our paper.
>
> **Relevance to NeurIPS community**
>
> Schrödinger bridges (SB) have attracted significant attention at NeurIPS as a key research theme at the intersection of diffusion models and optimal transport. Notably, several influential papers on SB have been published at NeurIPS, including Diffusion Schrödinger Bridge (NeurIPS 2021) [1] and Diffusion Schrödinger Bridge Matching (NeurIPS 2023) [2]. Our work reinforces the theoretical foundations that underpin this growing line of research.
>
> On the applied side, SB techniques have already been utilized in a wide range of real-world tasks. These include image restoration and transformation using I$^2$SB [3] (e.g., inpainting and super-resolution), unpaired image translation via UNSB [4] and SB Flow [5], domain adaptation through distribution alignment using maximum-likelihood SB solvers [6], and trajectory inference for single-cell RNA-seq data in the life sciences [6, 7].
>
> Taken together, the theoretical contributions of this work strengthen the foundation of SB-based methods and enhance our understanding of their practical applicability, making the results broadly relevant to the NeurIPS community.
> We will include the above related works in the final verstion.
>
>
> [1] De Bortoli, Valentin, et al. "Diffusion Schrödinger bridge with applications to score-based generative modeling." NeurIPS, 2021.
>
> [2] Shi, Yuyang, et al. "Diffusion Schrödinger bridge matching." NeurIPS, 2023.
>
> [3] Liu, Guan-Horng, et al. "I$^ 2$SB: Image-to-Image Schrödinger Bridge." arXiv:2302.05872, 2023.
>
> [4] Kim, Beomsu, et al. "Unpaired image-to-image translation via neural Schrödinger bridge." arXiv:2305.15086, 2023.
>
> [5] De Bortoli, V., Korshunova, I., Mnih, A., & Doucet, A. Schrödinger bridge flow for unpaired data translation. NeurIPS, 2024
>
> [6] Vargas, Francisco, et al. "Solving Schrödinger bridges via maximum likelihood." Entropy, 2021.
>
> [7] Chizat, Lénaïc, et al. "Trajectory inference via mean-field Langevin in path space." NeurIPS, 2022.

---

> > ### Comment · Reviewer_EHkZ · 2025-08-02
> > **thanks!**
> >
> > Thanks for clarification!

---

### Official Review · Reviewer_g9fY · 2025-07-03

**Clarity:** 4
**Significance:** 3
**Originality:** 3
**Rating:** 5
**Confidence:** 3

**Summary:**

This paper provides a theoretical study of a numerical scheme for approximating Schrodinger bridges between pairs of probability distributions.   More specifically,   it considers the Sinkhorn bridge approach of "Plug-in Estimation of Schrödinger Bridges" (Pooladian and Niles-Weed, 2024), and extends the theoretical guarantees.    In the original paper the authors assumed that one had access to the full source distribution, and only samples of the target.    In this paper, the novel contribution is the extension of this to the setting where only finitely many samples are available of both the source and the target (m and n respectively),   and provide an error bound on the Sinkhorn bridge iterates in TV distance.

Additionally, the authors note connections with other Schrodinger bridge approximations, and demonstrate that they can be recovered for various parameter choices and initialisations of the Sinkhorn Bridge algorithm.      The authors additionally perform a number of numerical experiments as empirical validation of the theory.

**Questions:**

I appreciate some of these questions are a bit open ended, I think they would add to the quality of the paper, but I already think this is a nice contribution as-is.

1. (The role of epsilon):   In practice, for Sinkhorn and related algorithms one needs to select the epsilon regularisation parameter wisely.   Depending on the setting, this is either selected through some cross-validation process, or sometimes annealed, or somestimes chosen as small as possible while maintaining numerical stability.    I can imagine that the sample size and epsilon are inter-related, particular due to the 'step-down' of the Holder exponent of the map (marginals -> coupling) in sinkhorn.    Can the authors comment on the numerical stability of the algorithm as epsilon->0.    Clearly, the bounds blow up as epsilon->0 in (14), etc, and this is expected, but in addition to this:  (i) is there some domain of validity, that can be elucidated through nuumerical experiment and (ii) do the bounds offer some recommendation on how to chose epsilon for given n, m?

2. (Experiments and d):  The authros  mention one limitation of the convergence bounds, namely the dependence on ambient dimension - something that is different from the original (Pooladian and Niles-Weed, 2024) bounds.    Would the authors be able to provide numerical experiments to demonstrate that the tightness or lack of tightness of the bounds in the case to provide support for their claim that "when the data are supported on a k -dimensional submanifold, the convergence behavior is expected to depend on the submanifold dimension k"  as they state in the conclusion?

3 (Stopping and guidance):  This might be asking too much, but can these bounds be applied to some sort of stopping rule for the iterates?   The bounds clearly suggest that there's a point at which there's no point in doing further sinkhorn iterations.   Can this be made explicit?  This would be  genuinely useful for practitioners.

**Ethical Concerns:**

["NO or VERY MINOR ethics concerns only"]

**Final Justification:**

The authors have provided additional details which substantially fill in some gaps of the paper, and make it more well-rounded.   Specifically, the authors addressed my queries about (i) the role of $\epsilon$, (ii) the dependence on dimension, and even provided a good stopping rule condition derived from their bounds.

Generally this paper is a generally strong theory paper, which I believe should be accepted.

**Limitations:**

Yes

**Quality:**

4

**Strengths And Weaknesses:**

Strengths:

* The paper fills an important gap in (Pooladian and Niles-Weed, 2024), extending the bounds in that paper to the more realistic setting where only samples are available for both the source and the target.
* The non-asymptotic bounds are provided for both the full estimator and the intermediate iterates.  This should, in practice, allow for practical guidance on sample size and number of iterations.
* The neat observation that many recent algorithms ([SF]^2 M, BM2, DSBM-IMF, LightSB, etc) are special cases of the Sinkhorn estimator is really neat - and means that the theoretical guarantees in this paper carry over.

Weaknesses:
* The limitation on the ambient dimension rather than the intrinsic dimension is a potential limitation, but one which the authors readily address.
* The numerical experiments - while highly elucidating - do not really provide any insight into how well the method scales in general - as they are simple examples, really focused on confirming the theory.
* The "coup de grace" that is missing from this paper, really whether the non-asymptotic results could be used to provide recommendations or guidance on how to choose the parameters, or even better provide some form of adaptive stopping (this might be asking too much given that the bounds are TV).
* As reductive as it sounds, the paper isn't a huge step, neither conceptually nor theoretically from (Pooladian and Niles-Weed, 2024) - but still very interesting and useful.

---

> ### Author Rebuttal · Authors · 2025-07-31
>
> We thank the reviewer for carefully reading our paper.
>
> **The role of epsilon**
>
> Revisiting the proof, we improved the exponential factor in equation (14) to $\exp\bigl(R^2(60+\varepsilon^{-1})\bigr),$
> and the following is written with that improvement in mind (to be reflected in the final version).
>
> The numerically unstable parts are the Sinkhorn Algorithm and the drift $b\_{m,n}^{(k)}$ with the instability being especially pronounced in the drift. This stems from the term appearing in equation (13)
> $$
> \exp\left(-\frac{\|Y_j - z\|_2^2}{2(1-t)\varepsilon}\right),
> $$
> where the denominator contains $(1-t)\varepsilon$; as $t$ approaches $1$, this induces sharp scaling. However, since the expression is in a softmax form, standard stabilization by subtracting the maximum of the exponent’s argument can be applied to ensure numerical stability.
>
> (i) To verify the empirical behavior, we ran the experiment for finding $\varepsilon$ that achieves target error $\delta=1.0$ under the same setup as in “Experiments (Statistical convergence)”, for given $m,n$. As predicted, the error drops sharply as $\varepsilon$ increases, then transitions to a regime of slower, asymptotic improvement—i.e., a steep improvement in $m\wedge n$ followed by a gradual change. The observed values were:
>
> | $m\wedge n$ | $\varepsilon$ |
> |--------------|----------------|
> | 10           | $2.684\times10^{8}$ |
> | 20           | $7.979\times10^{6}$ |
> | 30           | $3.200\times10^{1}$ |
> | 40           | $6.797\times10^{-1}$ |
> | 50           | $3.291\times10^{-1}$ |
> | 60           | $1.299\times10^{-1}$ |
> | 70           | $2.010\times10^{-1}$ |
> | 80           | $9.766\times10^{-4}$ |
>
> (ii) This phenomenon is supported theoretically: from equation (14), the $\varepsilon$ required to achieve target error $\delta$ satisfies
> $$
> \varepsilon \gtrsim \frac{R^2}{\log(\delta \cdot (m\wedge n))}.
> $$
> This indicates that when $m\wedge n$ is moderately large, error reduction proceeds rapidly, and thereafter improves only logarithmically.
>
>
> **Experiments and d**
>
> We conducted experiments to verify the dependence of the Sinkhorn bridge estimator on the intrinsic dimensionality. The experimental setup is as follows: the ambient dimension $d = 50$, the regularization strength $\varepsilon = 0.5$, and endpoint $\tau = 0.9$. We sampled the source distribution $\mu$ uniformly from the unit cube $[0,1]^d$, and the target distribution $\nu$ uniformly from the $d_\nu$‑dimensional unit sphere embedded in the ambient space $\mathbb{R}^d$: $\\{ x \in \mathbb{R}^{d_\nu} | \||x\||_2 = 1\\} \subset \mathbb{R}^d$.
>
> By varying $d_\nu$, we examined whether the convergence behavior of the estimation error with respect to sample sizes depends on the manifold dimension. For $d_\nu \in \\{5,\,10,\,20,\,30\\}$, we compared the time‑integrated mean squared error $\int_{0}^{0.9} \mathrm{MSE}_{\mathrm{sample}}(m,n,t)\,\mathrm{d}t,
> $ and observed that the error tends to increase as the manifold dimension grows, suggesting room for improvement.
>
> | $d_\nu$ | Error |
> | -------: | --------------------------------------------------------: |
> | $5$     | $0.031$                                                  |
> | $10$    | $0.138$                                                  |
> | $20$    | $0.562$                                                  |
> | $30$    | $1.178$                                                  |
>
> **Stopping and guidance**
>
> Considering the decomposition of TV-distance, the number of iterations $k$ satisfying the following inequality is sufficient: $\mathbb{E}[ TV^2(P\_{\infty,\infty}^{\*,[0,\tau]}, P\_{m,n}^{\*,[0,\tau]})] \geq \mathbb{E}\bigl[TV^2(P\_{m,n}^{\*,[0,\tau]}, P\_{m,n}^{k,[0,\tau]})]$. This implies $k \geq C\frac{\log\bigl(A/n + B/(m\wedge n)\bigr)}
>      {\log\bigl(\tanh(R^2/\varepsilon)\bigr)}$ where $A = \frac{1}{(1-\tau)^{d+1} \varepsilon^{d-1}R^2}$ and $B = \frac{C_{d}R^{5(d+1)}}{(1-\tau)\tau\varepsilon} e^{R^2\bigl(60 + \varepsilon^{-1}\bigr)}$. Since $\log(\tanh(R^2/\varepsilon))<0$, the numerator $\log\bigl(A/n + B/(m\wedge n)\bigr)$ must also be negative for the right-hand side to be meaningful. This in turn implies that
> $m \wedge n \approx \max\{A, B\}
> $ samples are needed.
>
> This prediction is also confirmed by Figure 3. In particular, $b_{m,n}^{(1)}$ obtained after a single Sinkhorn iteration, already transports points to their nearest neighbors and captures the bulk of $\nu_n$.
>
> The only remaining discrepancy with the optimal drift $b_{m,n}^*$ lies in the mismatch of the marginal distributions, similar to the case of entropic optimal transport. As a result, a sufficiently large $k$ becomes necessary only when the empirical distributions $\mu_m$ and $\nu_n$ are already close to the true distributions $\mu$ and $\nu$ i.e., when $m$ and $n$ are sufficiently large.

---

> > ### Comment · Reviewer_g9fY · 2025-08-03
> > **Thank you**
> >
> > Thank you for these additional details, which are very illuminating.  I strongly encourage the authors to integrate these points into your final version.

---

### Decision · Program_Chairs · 2025-09-17

**Decision:**

Accept (poster)

**Comment:**

This work establishes finite-sample error bounds for Sinkhorn bridge iterations in the two-sample Schrödinger bridge setting, providing theoretical guarantees, practical guidance on sample sizes and iteration numbers, and unifying several existing methods under a common framework.

Strength: This paper extends the results of Pooladian and Niles-Weed (2024) to the more practical/realistic two-sample estimation setting.

Weakness: Some limited empirical results.

The paper makes an important contribution to the recent literature on analysis of Sinkhorn algorithms. All reviewers unanimously recommend acceptance, which the AC also concurs with in their own reading.